# Drivers of long-lasting insecticide-treated net utilisation and parasitaemia among under-five children in 13 States with high malaria burden in Nigeria

Perpetua Uhomoibhi[1], Chukwu Okoronkwo[1], IkeOluwapo O. Ajayi[2,3], Olugbenga Mokuolu[1,4], Ibrahim Maikore[5], Adeniyi Fagbamigbe[2], Joshua O. Akinyemi[2,6]*, Festus Okoh[1], Cyril Ademu[1], Issa Kawu[1], Jo-Angeline Kalambo[7], James Ssekitooleko[7]

1 National Malaria Elimination Programme, Federal Ministry of Health, Abuja, Nigeria, 2 Department of Epidemiology and Medical Statistics, College of Medicine, University of Ibadan, Ibadan, Nigeria, 3 Epidemiology and Biostatistics Research Unit, Institute for Advanced Medical Research and Training, College of Medicine, University of Ibadan, Ibadan, Nigeria, 4 Department of Paediatrics, University of Ilorin Teaching Hospital, Ilorin, Nigeria, 5 World Health Organization Country Office, Abuja, Nigeria, 6 Infectious Diseases Institute, College of Medicine, University of Ibadan, Ibadan, Nigeria, 7 High Impact Africa 1 Department, Global Fund to Fight AIDS, TB and Malaria, Geneva, Switzerland

* joakinyemi@com.ui.edu.ng, odunjoshua@gmail.com

**Data Availability Statement:** The data underlying the results presented in the study are available from https://www.dhsprogram.com/.

## Abstract

### Background

Although Nigeria has made some progress in malaria control, there are variations across States. We investigated the factors associated with utilisation of long-lasting insecticide-treated net (LLIN) and parasitaemia among under-five children in 13 States with high malaria burden.

### Method

Data from the 2015 Nigeria Malaria Indicator Survey and 2018 Demographic and Health Survey were obtained and analysed. The 2015 and 2018 data were compared to identify States with increase or reduction in parasitaemia. Analysis was done for all the 13 study States; four States with increased parasitaemia and nine States with reduction. Random-effects logit models were fitted to identify independent predictors of LLIN utilisation and parasitaemia.

### Results

LLIN was used by 53.4% of 2844 children, while parasitaemia prevalence was 26.4% in 2018. Grandchildren (AOR = 5.35, CI: 1.09–26.19) were more likely to use LLIN while other relatives (AOR = 0.33, CI: 0.11–0.94) were less likely compared to children of household-heads. LLIN use was more common in children whose mother opined that only weak children could die from malaria (AOR = 1.83, CI: 1.10–3.10). Children whose mothers obtained net from antenatal or immunisation clinics (AOR = 5.30, CI: 2.32–12.14) and campaigns

**Funding:** The author(s) received no specific funding for this work.

**Competing interests:** The authors have declared that no competing interests exist.

(AOR = 1.77, CI: 1.03–3.04) were also more likely to use LLIN. In contrast, LLIN utilisation was less likely among children in female-headed households (AOR = 0.51, CI: 0.27–0.99) and those in poor-quality houses (AOR = 0.25, CI: 0.09–0.72).

Children aged 24–59 months compared to 0–11 months (AOR = 1.78, CI: 1.28–2.48), those in whom fever was reported (AOR = 1.31, CI: 1.06–1.63) and children of uneducated women (AOR = 1.89, CI: 1.32–2.70) were more likely to have parasitaemia. The likelihood of parasitaemia was higher among children from poor households compared to the rich (AOR = 2.06, CI: 1.24–3.42). The odds of parasitaemia were 98% higher among rural children (AOR = 1.98, CI: 1.37–2.87).

## Conclusion

The key drivers of LLIN utilisation were source of net and socioeconomic characteristics. The latter was also a key factor associated with parasitaemia. These should be targeted as part of integrated malaria elimination efforts.

## Introduction

Malaria remains a persistent threat to the world, especially in sub-Saharan Africa and South Asia, where it is endemic. This is despite huge investments in the production and distribution of Long-Lasting Insecticide-treated Net (LLINs) and the deployment of effective antimalaria drugs, which development partners mostly support. According to the 2021 World Malaria Report, there were about 241 million malaria cases in 2020 across 85 endemic countries, with a 14 million additional cases compared to 2019. The increase were partly attributed to COVID-19 disruptions [1]. The estimated number of malaria deaths stood at 627,000 in 2020, showing an upwardtrend compared to 2019 [1]. Children under five years of age and pregnant women are the most vulnerable groups affected by malaria. In 2019, the under-fives accounted for 61% (266 000) of all malaria deaths worldwide. Over 90% of these cases occur in sub-Saharan Africa, and Nigeria contributes the most cases of any country globally, at 23% [2].

Every year, about 110 million clinically diagnosed malaria cases and 300,000 malaria-related childhood deaths occur in Nigeria [2]. Ninety-five percent of these are due to *Plasmodium falciparum*, the most predominant malaria parasite in the country. The disease also causes substantial economic losses in out-of-pocket payment, prevention costs, and loss of person-hours [3].

Despite its malaria burden, Nigeria has made some progress in curtailing the menace. For instance, the prevalence of malaria in Nigeria decreased from 27% in 2015 to 23% in 2018, while the coverage of LLINs increased from 42% to 69% [4]. This progress is a direct result of huge support and investment by different stakeholders and multilateral partners such as the Global Fund grant, President's Malaria Initiative (PMI) and other organisations. The Global Fund (GF) grant programme is based on the National Malaria Strategic Plan (NMSP), which is in line with the Global Technical Strategy (GTS) targets towards scaling up malaria interventions and health systems strengthening. The grant seeks to contribute to the achievement of Nigeria's long-term goal of eliminating malaria as a public health problem through universal coverage of key interventions.

Aside the aggregate reduction in malaria prevalence between 2015 and 2018, there are significant variations in prevalence across the different states. For example, 4 of 13 states with

high malaria burden had increased prevalence over the period despite a high level of net utilisation and a near-full implementation of preventive and curative services. These imply a need to take a deeper look into the factors driving the prevalence of malaria in these states since this is not completely explainable by the distribution of LLINs only. This brings about the questions: what could be the drivers of high levels of parasitaemia and preventive methods utilisation, especially the LLIN? These are the questions that led to the search for the determinants of LLIN use and malaria parasitaemia in 13 Nigerian States with high malaria burden.

A wide range of factors have been reported in the literature to be associated with malaria parasitaemia among under-five children. These include factors related to children, mothers, households, the community, and the health system. For instance, parasitaemia was reported to be more prevalent in older children than among infants [5]. In addition, children belonging to women with secondary/higher education and those from rich households were reported to have lower risks [6]. It has also been shown that household environmental and community characteristics such as sanitation, quality of housing materials, bushy environment, presence of livestock in the household are risk factors [7–10]. The magnitude of these risk factors varies across settings and is moderated by health systems and malaria prevention programmes. This is a typical scenario in the Nigerian context with a diverse ecological, socioeconomic and health system profile across different States and geopolitical zones. To generate evidence that can be used to design targeted interventions, this study focused on 13 States with a high prevalence of malaria. Four of these states had increased parasitaemia between 2015 and 2018, while the remaining nine experienced a decrease [4,11]. It is necessary to unravel the correlates of parasitaemia in these two categories of States so that malaria elimination programmes can be better refined.

Therefore, this study aimed to answer the question, "What are the individual, household and community level drivers of LLIN utilisation and parasitaemia among under-5 children in the 13 states with high malaria burden in Nigeria? We also addressed the same question in States with increased parasitaemia and those with a reduction between 2015 and 2018.

## Methods

### Description of data sources

This study involves analysis of secondary data obtained from the 2015 Nigeria Malaria Indicator Survey (NMIS) and the 2018 Nigeria Demographic and Health Survey (NDHS). These are nationally representative datasets. The 2015 data set was used to categorize states as having high or low malaria burden, and to compare with 2018 data for indicating increased or reduced parasitaemia.

The 2015 Nigeria Malaria Indicator Survey (NMIS) was conducted in all Nigerian States and the Federal Capital Territory between October and November 2015 [11]. All women aged 15–49 years old who were either permanent residents of the households or visitors in the households on the night before the survey were eligible to be interviewed. In addition, all children aged 6–59 months were eligible to be tested for malaria and anaemia. Nationally representative samples of over 7745 households in 329 clusters were sampled. This sample size was selected to provide power to estimate key survey indicators for the country and the six geopolitical zones. A more detailed description of survey design and microscopy procedures can be found in the NMIS 2015 report [11].

Similarly, in the NDHS 2018 conducted between August and December, all women aged 15–49 years old who were either permanent residents of the households or visitors on the night before the survey were eligible to be interviewed. In addition, all children aged 6–59

months were eligible to be tested for malaria and anaemia. A detailed description of the sample design and other profiles of the NDHS 2018 is available in the full report [4].

## Sampling techniques in NMIS 2015 and NDHS 2018

The data for NMIS 2015 and 2018 NDHS were both obtained from stratified samples selected in two stages. Sampling frames (enumeration areas) were based on the Population and Housing Census of the Federal Republic of Nigeria (NPHC) conducted in 2006. The Enumeration Areas (EAs) constituted the primary sampling unit (PSU). Stratification was achieved by classifying the 36 states and the Federal Capital Territory into urban and rural areas. Samples were selected independently in every stratum via a two-stage selection. Probability proportional to size selection was used during the first stage of sampling. Household listing and numbering were done to generate a sampling frame for the second stage. In the second stage, systematic sampling was done to select 30 households per EAs.

## Data collection

Data were collected by trained interviewers who visited selected households to enrol eligible respondents. Questionnaires were administered to household heads and women aged 15–49 years. In addition, blood samples were collected to screen under-five children for parasitaemia and anaemia. Malaria testing was based on a rapid diagnostic test kit and microscopy. However, analysis in this study was based on microscopy results.

## Study population

Our population of interest in this study were under-five children and their mothers/caregivers in thirteen states with high malaria burden based on the NDHS 2018. These were Adamawa, Delta, Gombe, Jigawa, Kaduna, Kano, Katsina, Kwara, Niger, Ogun, Osun, Taraba, and Yobe.

## Variables

The outcome variables were utilisation of LLIN, and malaria microscopy result confirming positivity or otherwise in under-five children. Two conceptual frameworks based on insight from the literature guided variables selection for analysis (Figs 1 & 2). We explored a set of explanatory variables measured at the individual (child and mother), household, and community levels. A summary of these variables is presented in Table 1.

Cluster (PSU) used during sample selection was adopted as a proxy for the community. Therefore, the community-level variables were derived from existing individual and household variables as follows:

- Community illiteracy level: the proportion of children whose mothers have no formal education in the cluster

- Community poverty level: the proportion of children from households in the lowest two wealth quintiles within the cluster. These proportions were categorised into two levels (low and high)using the 50th percentile cut-off to allow for non-linear effects and offer useful results for policy decisions. Similar procedures have been used in the literature [12–14].

- Community disadvantage: This was derived using principal component analysis to aggregate the neighbourhood factors such as type of residence, formal education level and household wealth quintile. Standardised scores with zero mean and one standard deviation were generated and categorised into 3 (low, medium and high).

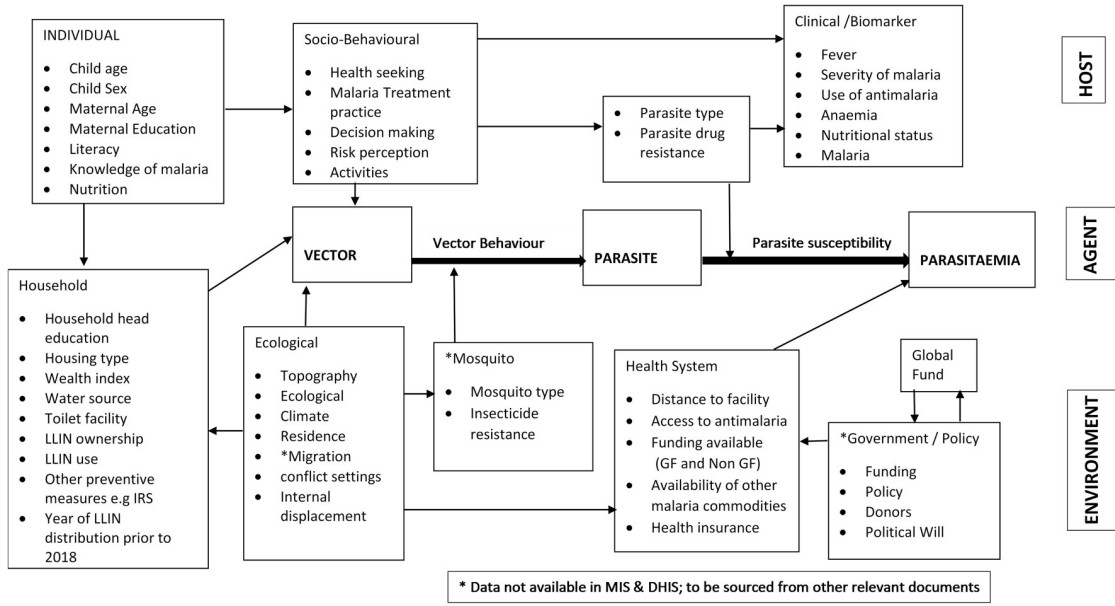

**Fig 1. Conceptual framework on the driving forces that influence the change of parasitaemia in 13 States with high malaria burden.**

**Data analysis.** Based on the comparison of parasitaemia prevalence between NMIS 2015 and NDHS 2018, states were classified into two groups- those with an increase in parasitaemia and those with a reduction. After that, we analysed 2018 NDHS data and presented results for (i) all the 13 States; (ii) Four States with increased parasitaemia (Gombe, Jigawa, Kano and Ogun) and (iii) Nine States with a reduction in parasitaemia (Adamawa, Delta, Kaduna, Katsina, Kwara, Niger, Osun, Taraba, and Yobe).

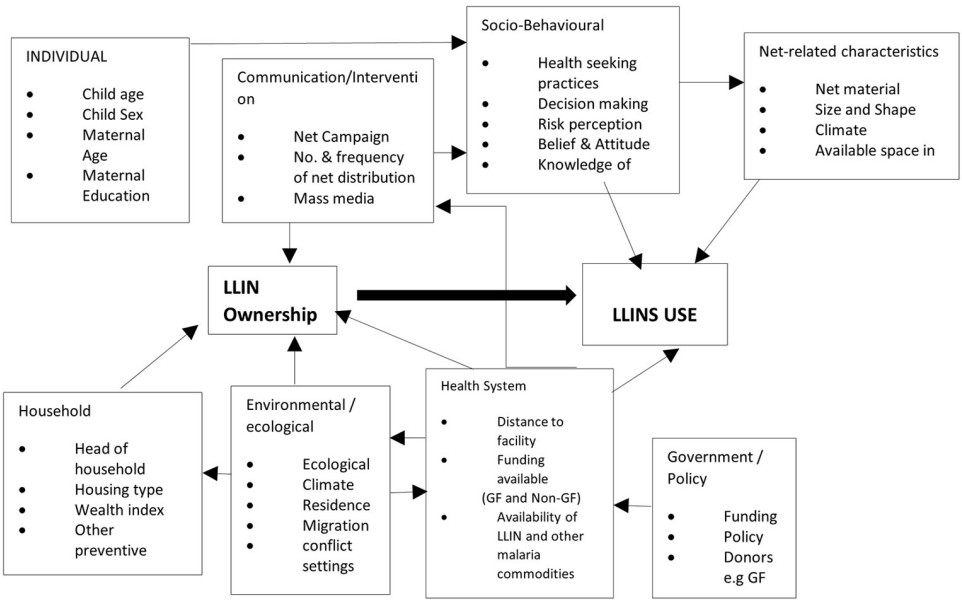

**Fig 2. Conceptual framework on the driving forces that influence the change in use of LLIN in 13 States with high malaria burden.**

**Table 1. List of explanatory variables.**

| Individual level factors | Household level factors | Community-level factors |
|---|---|---|
| Child sex [male; female] | Age of household head [<25;25–34; 35–49; >=50] | Altitude [low land; plain land; high land] |
| Child age in months [0–11; 12–23; 24–59] | Sex of household head [Male; Female] | Place of residence [rural; urban] |
| Relationship to head of household [child; a grandchild; other relatives] | Wealth index [poor; middle; rich] | Community illiteracy level [Low; High] |
| Nutritional status Stunting [Yes or No] Wasting [Yes or No] | Quality of housing material [Good; average; poor] | Community poverty level [Low; High] |
| Use of LLIN (Child slept under net) [Yes; No] | Type of toilet facility [Improved; Not improved] | Community disadvantage [Low; Medium; High] |
| Child had a fever in the last two weeks [Yes; No] | Source of drinking water [Improved; Not improved] | |
| Treatment of fever in the last two weeks [Yes; No] | Presence of livestock around the house [Yes; No] | |
| Maternal age in years [<25; 25–34; 35–49] | Ownership of radio/TV [Yes; No] | |
| Maternal education [none; primary; secondary/ higher] | | |
| Maternal occupation [not working; white-collar job; Sales; Agriculture/manual work] | | |
| Maternal involvement in household decision making [poor; average; good;] | | |
| Exposure to radio/television [not at all; less than once a week; at least once a week] | | |

Weighted analysis was conducted for each outcome variable (LLIN use and parasitaemia). First, frequencies and percentages were used to summarise the outcome and explanatory variables. Next, random-effect logit models were fitted to investigate the association between each explanatory variables and the outcomes. Factors with p-values<0.05 were entered in a multivariable model to identify the independent determinants of LLIN use and parasitaemia. Finally, we implemented random-effect models in which individual mother/child pair was nested in clusters (communities). This allowed us to adjust for the complex sampling procedure used for the survey while controlling for the clustering of observations within communities.

**Ethical consideration.** The survey protocols for the 2015 NMIS (NHREC/01/01/2007-11/ 05/2015) and 2018 NDHS were approved by the National Health Research Ethics Committee of the Federal Ministry of Health (NHREC), as well as the Review Board of ICF International. In addition, written informed consent was obtained from respondents and parents of the under-five children prior to administration of questionnaire and blood sample collection from the children. Participation was entirely voluntary. We also obtained formal approval (Author-Letter_152682 dated February 23[rd], 2021) from Measure DHS for the analyses reported in this study.

## Results

### Background characteristics of children and their mother/caregiver

The distribution of the children, mothers, household, and community characteristics according to prevalence of parasitaemia and LLIN use in the study States are presented in Table 2.

**Table 2. Distribution of the children characteristics in the study States, NDHS 2018.**

| Variables/Categories | All 13 States | | | | States with Increased parasitaemia | | | | States with parasitaemia reduction | | | |
|---|---|---|---|---|---|---|---|---|---|---|---|---|
| | n | % | Para (%) | LLIN (%) | n | % | Para (%) | LLIN (%) | n | % | Para (%) | LLIN (%) |
| **Sex** | | | | | | | | | | | | |
| Male | 1,445 | 51.7 | 27.6 | 54.1 | 512 | 50.5 | 31.5 | 60.2 | 933 | 52.4 | 25.6 | 51.2 |
| Female | 1,399 | 48.3 | 25.1 | 52.7 | 507 | 49.5 | 29.7 | 57.7 | 892 | 47.6 | 22.7 | 50.1 |
| **Child's Age (Months)** | | | | | | | | | | | | |
| 0–11 | 346 | 12.7 | 21.1 | 56.3 | 124 | 12.8 | 18.1 | 59.9 | 222 | 12.7 | 22.6 | 54.6 |
| 12–23 | 683 | 23.4 | 21.2 | 54.0 | 243 | 23.0 | 26.1 | 60.1 | 440 | 23.6 | 18.7 | 50.9 |
| 24–59 | 1,815 | 63.9 | 29.3 | 52.1 | 652 | 64.2 | 34.7 | 58.2 | 1,163 | 63.7 | 26.6 | 49.0 |
| **Anaemia Level** | | | | | | | | | | | | |
| Severe | 91 | 3.0 | 67.5 | 54.5 | 38 | 3.6 | 83.0 | 61.8 | 53 | 2.7 | 57.1 | 49.0 |
| Moderate | 1,104 | 38.3 | 39.2 | 54.8 | 447 | 44.5 | 40.7 | 64.3 | 657 | 35.1 | 38.1 | 48.5 |
| Mild | 717 | 25.2 | 19.9 | 52.5 | 249 | 23.9 | 23.5 | 57.1 | 468 | 25.9 | 18.3 | 50.3 |
| Not Anaemia | 932 | 33.5 | 12.9 | 51.8 | 285 | 28.0 | 13.8 | 52.4 | 647 | 36.3 | 12.5 | 51.5 |
| **Had Fever In Last 2 Weeks** | | | | | | | | | | | | |
| No | 1,983 | 70.8 | 23.3 | 52.3 | 731 | 72.6 | 26.0 | 56.2 | 1,252 | 70.0 | 21.9 | 50.2 |
| Yes | 861 | 29.2 | 33.9 | 56.5 | 288 | 27.4 | 42.9 | 66.7 | 573 | 30.0 | 29.7 | 51.7 |
| **Took antimalaria (ACT)** | | | | | | | | | | | | |
| No | **735** | 78.5 | 34.4 | 52.9 | 227 | 73.4 | 32.9 | 62.2 | 506 | 80.7 | 35.1 | 48.4 |
| Yes | **202** | 21.5 | 19.1 | 57.7 | 83 | 26.6 | 22.0 | 48.1 | 121 | 19.3 | 17.8 | 62.0 |
| **Stunting** | | | | | | | | | | | | |
| No | 1,514 | 52.4 | 21.6 | 49.5 | 470 | 45.1 | 23.8 | 52.6 | 1,044 | 56.1 | 20.8 | 48.1 |
| Yes | 1,330 | 47.6 | 31.6 | 58.4 | 549 | 54.9 | 36.2 | 65.3 | 781 | 43.9 | 28.6 | 54.3 |
| **Wasting** | | | | | | | | | | | | |
| No | 2,649 | 92.9 | 26.0 | 52.9 | 939 | 92.3 | 30.2 | 58.6 | 1,710 | 93.2 | 23.8 | 50.0 |
| Yes | 195 | 7.1 | 31.9 | 61.4 | 80 | 7.7 | 35.0 | 63.3 | 115 | 6.8 | 30.0 | 60.4 |
| **Child Slept under LLIN** | | | | | | | | | | | | |
| No | 1,416 | 46.1 | 26.6 | | 440 | 40.0 | 30.8 | | 976 | 49.3 | 24.9 | |
| Yes | 1,428 | 53.9 | 26.2 | | 579 | 60.0 | 30.5 | | 849 | 50.7 | 23.6 | |
| **Relationship to Household Head** | | | | | | | | | | | | |
| Child | 2,652 | 94.2 | 26.0 | 54.8 | 947 | 94.1 | 30.0 | 60.6 | 1,705 | 94.3 | 24.0 | 51.9 |
| Grandchild | 132 | 3.8 | 30.2 | 30.0 | 40 | 3.0 | 27.2 | 27.2 | 92 | 4.3 | 31.3 | 30.9 |
| Other Relative | 60 | 1.9 | 36.3 | 41.1 | 32 | 2.9 | 51.7 | 45.0 | 28 | 1.4 | 19.4 | 36.4 |

Para: Parasitaemia; ACT: Artemisinin Combination Therapy; LLIN: Long Lasting Insecticide-treated Net.

Overall, about 52% of the under-five children were males, while 23.4% were aged 12–23 months and about 64% were aged 24–59 months. Fever in the past two weeks before data collection was reported in about 3 out of 10 children. In addition, close to half of all children were stunted (47.6%), but the prevalence of wasting was far lesser (7.1%).

Maternal, household and community characteristics are summarised in Table 3. The majority (55.2%) of under-five children had mothers with no formal education, while 30.3% had mothers with secondary education. About half belonged (49%) to women aged 25–34 years. Most children had mothers with poor involvement in basic household decisions (63%).

About 6 out of 10 children lived in households with livestock or farm animals around the house. Household size distribution showed that 75% had at least five members, and only 2.2% used health insurance. Forty-four percent of children dwell in houses with totally improved quality while 11.0% had unimproved houses. Overall, 36.4% were residents in urban areas.

**Table 3. Distribution of the mothers', household, and community characteristics in the study States, NDHS 2018.**

| Variables/Categories | All 13 States | | | | States with Increased parasitaemia | | | | States with parasitaemia reduction | | | |
|---|---|---|---|---|---|---|---|---|---|---|---|---|
| | n | % | Para | LLIN | n | % | Para | LLIN | n | % | Para | LLIN |
| **Mother's Highest Educational Level** | | | | | | | | | | | | |
| No Education | 1,570 | 55.2 | 34.0 | 56.8 | 595 | 58.1 | 39.9 | 65.0 | 975 | 53.7 | 30.6 | 52.3 |
| Primary | 422 | 14.5 | 25.8 | 52.8 | 138 | 13.3 | 30.8 | 56.3 | 284 | 15.1 | 23.5 | 51.3 |
| Secondary and higher | 852 | 30.3 | 12.9 | 47.2 | 286 | 28.6 | 11.6 | 46.9 | 566 | 31.2 | 13.5 | 47.3 |
| **Mother's Age** | | | | | | | | | | | | |
| 15–24 | 659 | 23.4 | 26.9 | 61.0 | 241 | 22.8 | 33.3 | 67.0 | 418 | 23.7 | 23.7 | 57.7 |
| 25–34 | 1,393 | 48.6 | 26.4 | 52.4 | 482 | 47.6 | 27.7 | 57.1 | 911 | 49.1 | 25.7 | 50.3 |
| 35–49 | 792 | 28.0 | 25.9 | 48.4 | 296 | 29.5 | 33.1 | 54.7 | 496 | 27.2 | 22.0 | 45.0 |
| **Net from Campaign** | | | | | | | | | | | | |
| No | 358 | 21.5 | 20.6 | 89.5 | 140 | 20.8 | 26.2 | 85.3 | 218 | 22.0 | 16.9 | 92.8 |
| Yes | 1,241 | 78.5 | 28.2 | 91.0 | 570 | 79.2 | 32.9 | 81.8 | 671 | 78.0 | 24.8 | 97.1 |
| **Household Head's Age** | | | | | | | | | | | | |
| <25 | 70 | 2.7 | 28.0 | 54.7 | 29 | 3.1 | 32.3 | 54.7 | 41 | 2.4 | 25.2 | 54.8 |
| 25–34 | 744 | 25.1 | 22.9 | 56.8 | 276 | 26.5 | 25.4 | 60.9 | 468 | 24.3 | 21.5 | 54.5 |
| 35–49 | 1,371 | 49.1 | 26.4 | 54.2 | 469 | 47.1 | 30.4 | 58.2 | 902 | 50.1 | 24.4 | 52.3 |
| >=50 | 658 | 23.2 | 30.0 | 47.8 | 245 | 23.3 | 36.7 | 59.0 | 413 | 23.1 | 26.5 | 42.4 |
| **Media Access** | | | | | | | | | | | | |
| No | 611 | 19.1 | 33.6 | 56.3 | 205 | 18.4 | 41.9 | 68.0 | 406 | 19.4 | 29.6 | 51.3 |
| Yes | 2,233 | 80.9 | 24.7 | 52.6 | 814 | 81.6 | 28.1 | 56.8 | 1,419 | 80.6 | 22.9 | 50.5 |
| **Radio/TV exposure** | | | | | | | | | | | | |
| No | 1,227 | 41.5 | 31.8 | 58.7 | 375 | 34.1 | 40.6 | 66.0 | 852 | 45.4 | 28.4 | 56.0 |
| < Once a wk | 609 | 21.9 | 25.5 | 47.8 | 276 | 30.4 | 28.6 | 47.6 | 333 | 17.5 | 22.7 | 48.0 |
| > Once a wk | 1,008 | 36.5 | 20.7 | 50.6 | 368 | 35.4 | 22.6 | 61.9 | 640 | 37.1 | 19.8 | 45.2 |
| **Decision-making involvement** | | | | | | | | | | | | |
| Poor | 1,752 | 63.0 | 26.6 | 57.3 | 655 | 62.2 | 30.7 | 64.8 | 1,097 | 63.5 | 24.5 | 53.7 |
| Average | 371 | 13.5 | 32.8 | 49.2 | 179 | 20.1 | 33.0 | 53.3 | 192 | 10.1 | 32.6 | 44.9 |
| Good | 721 | 23.5 | 22.1 | 45.1 | 185 | 17.7 | 27.4 | 44.8 | 536 | 26.4 | 20.3 | 45.2 |
| **Owns Livestock, Herds Or Farm Animals** | | | | | | | | | | | | |
| No | 1,128 | 40.4 | 19.4 | 46.1 | 376 | 39.4 | 20.3 | 47.6 | 752 | 41.0 | 19.0 | 45.4 |
| Yes | 1,716 | 59.6 | 31.1 | 58.2 | 643 | 60.6 | 37.3 | 65.7 | 1,073 | 59.0 | 27.8 | 54.2 |
| **Number of De Facto Members** | | | | | | | | | | | | |
| <5 | 679 | 24.0 | 19.5 | 55.1 | 220 | 22.1 | 24.4 | 58.0 | 459 | 25.0 | 17.3 | 53.7 |
| >=5 | 2,165 | 76.0 | 28.6 | 52.9 | 799 | 77.9 | 32.3 | 59.3 | 1,366 | 75.0 | 26.5 | 49.7 |
| **Number of Dejure U5C** | | | | | | | | | | | | |
| 1 | 700 | 23.7 | 21.1 | 50.2 | 224 | 20.8 | 23.0 | 56.0 | 476 | 25.1 | 20.3 | 47.7 |
| 2 | 1,163 | 40.9 | 24.8 | 55.2 | 405 | 41.0 | 28.5 | 56.8 | 758 | 40.9 | 22.9 | 54.4 |
| >=3 | 966 | 35.4 | 31.8 | 54.0 | 385 | 38.3 | 36.9 | 63.9 | 581 | 34.0 | 28.8 | 48.7 |
| **Number of Mosquito Bed Nets** | | | | | | | | | | | | |
| None | 768 | 23.3 | 22.4 | 0.0 | 157 | 13.5 | 21.6 | 0.0 | 611 | 28.4 | 22.6 | 0.0 |
| One | 458 | 17.2 | 24.8 | 59.4 | 135 | 12.9 | 26.5 | 63.5 | 323 | 19.4 | 24.2 | 58.1 |
| Two or more | 1,618 | 59.5 | 28.4 | 73.7 | 727 | 73.5 | 33.0 | 69.9 | 891 | 52.2 | 25.1 | 76.3 |
| **Source of Drinking Water** | | | | | | | | | | | | |
| Not Improved | 1,203 | 39.3 | 30.3 | 51.9 | 408 | 38.7 | 33.6 | 54.7 | 795 | 39.6 | 28.6 | 50.6 |
| Improved | 1,641 | 60.7 | 23.9 | 54.5 | 611 | 61.3 | 28.7 | 61.7 | 1,030 | 60.4 | 21.4 | 50.7 |
| **Type of Toilet Facility** | | | | | | | | | | | | |
| Improved | 1,523 | 53.2 | 21.1 | 49.6 | 581 | 56.6 | 23.2 | 50.7 | 942 | 51.4 | 19.9 | 48.9 |

*(Continued)*

**Table 3.** (Continued)

| Variables/Categories | All 13 States | | | | States with Increased parasitaemia | | | | States with parasitaemia reduction | | | |
|---|---|---|---|---|---|---|---|---|---|---|---|---|
| | n | % | Para | LLIN | n | % | Para | LLIN | n | % | Para | LLIN |
| Not Improved | 1,321 | 46.8 | 32.4 | 57.6 | 438 | 43.4 | 40.2 | 69.5 | 883 | 48.6 | 28.8 | 52.4 |
| **Have Health Insurance** | | | | | | | | | | | | |
| No | 2,781 | 97.8 | 26.9 | 53.5 | 990 | 97.1 | 31.3 | 58.9 | 1,791 | 98.1 | 24.6 | 50.8 |
| Yes | 63 | 2.2 | 4.4 | 49.8 | 29 | 2.9 | 7.5 | 61.0 | 34 | 1.9 | 2.0 | 41.5 |
| **Household Wealth Tertiles** | | | | | | | | | | | | |
| Poorest | 1,157 | 38.4 | 37.2 | 57.7 | 488 | 46.0 | 40.7 | 67.0 | 669 | 34.4 | 34.8 | 51.7 |
| Middle | 985 | 35.1 | 25.1 | 53.8 | 263 | 25.2 | 32.2 | 59.4 | 722 | 40.3 | 22.8 | 51.9 |
| Richest | 702 | 26.5 | 12.4 | 46.3 | 268 | 28.7 | 13.0 | 44.5 | 434 | 25.3 | 12.0 | 47.3 |
| **Housing quality** | | | | | | | | | | | | |
| Totally Improved | 1,171 | 44.0 | 19.3 | 46.4 | 337 | 35.5 | 20.3 | 49.5 | 834 | 48.4 | 18.9 | 45.2 |
| Some Improved | 1,284 | 44.9 | 31.7 | 59.2 | 545 | 53.3 | 35.6 | 62.3 | 739 | 40.6 | 29.1 | 57.1 |
| Totally Unimproved | 385 | 11.1 | 33.1 | 57.1 | 136 | 11.2 | 39.9 | 71.8 | 249 | 11.1 | 29.6 | 50.2 |
| **Location** | | | | | | | | | | | | |
| Urban | 981 | 36.4 | 14.3 | 48.8 | 334 | 36.8 | 16.5 | 57.6 | 647 | 36.2 | 13.2 | 44.5 |
| Rural | 1,863 | 63.6 | 33.3 | 55.7 | 685 | 63.2 | 38.8 | 59.6 | 1,178 | 63.8 | 30.5 | 53.7 |
| **Community Poverty** | | | | | | | | | | | | |
| Low | 1,337 | 47.6 | 24.8 | 55.8 | 365 | 33.1 | 27.4 | 67.3 | 972 | 55.1 | 24.0 | 52.1 |
| High | 1,507 | 52.4 | 27.8 | 51.2 | 654 | 66.9 | 32.2 | 54.4 | 853 | 44.9 | 24.5 | 48.8 |
| **Community Illiteracy** | | | | | | | | | | | | |
| Low | 1,317 | 46.4 | 16.6 | 52.3 | 485 | 46.3 | 16.4 | 56.1 | 832 | 46.5 | 16.7 | 50.5 |
| High | 1,527 | 53.6 | 34.9 | 54.3 | 534 | 53.7 | 42.8 | 61.2 | 993 | 53.5 | 30.8 | 50.8 |
| **Community SES Disadvantage** | | | | | | | | | | | | |
| Low | 663 | 24.4 | 13.9 | 38.2 | 212 | 23.3 | 10.3 | 36.8 | 451 | 25.0 | 15.6 | 38.9 |
| Medium | 893 | 29.5 | 23.7 | 55.1 | 262 | 23.4 | 27.4 | 61.5 | 631 | 32.7 | 22.4 | 52.7 |
| High | 1,288 | 46.0 | 34.8 | 60.0 | 545 | 53.3 | 40.9 | 66.7 | 743 | 42.3 | 30.8 | 55.8 |
| **Total** | **2,844** | | **26.4** | **53.4** | **1,019** | | **30.6** | **59.0** | **1,825** | | **24.2** | **50.6** |

Slightly more than half (52.4%) of children resided in communities with a high level of illiteracy, while 46% were from settings with the highest socioeconomic disadvantage.

The distribution pattern for most of the variables was similar across states with increased parasitaemia and those with reduction except for household wealth tertile, housing quality, and community poverty. For instance, the percentage of children in poorest wealth quintile in States with increased parasitaemia (46.0%) was higher than those of States with parasitaemia reduction (34.4%). In comparison, the proportion in the middle tertile was higher in the latter (40.3%) than in the former (25.5%). Similarly, the percentage of under-fives in communities with the highest socioeconomic disadvantage was higher in States with increased parasitaemia (53.3%) than those with reduction (42.3%).

**Utilisation of LLIN.** The 2018 data showed that LLIN was used for 53.4% of under-five children in the 13 study States (Table 2). LLIN utilisation was evenly distributed across many background characteristics apart from variables such as wasting, stunting, education, age of mother, place of residence and community disadvantage (Tables 2 and 3). For instance, utilisation was higher in stunted and wasted children compared to those without these conditions (Table 2). Further, LLIN utilisation declined with maternal educational attainment while it increased with community disadvantaged (Table 3). Lastly, LLIN utilisation was lower in urban (48.8%) than rural areas (55.7%).

**Prevalence of parasitaemia.** In 2018 NDHS, the overall prevalence of parasitaemia among under-five children was 26.4%. The level was highest among children aged 24–59 months. It was also higher among those with fever in the past two weeks (Table 2). Similarly, parasitaemia prevalence increased with the severity of anaemia. Further, it was higher among stunted children (Yes- 31.6%, No– 21.6%). In terms of maternal education, it ranged from 34% in children whose mothers had no formal education to 12.9% in those with secondary/ higher education. The prevalence of parasitaemia was higher among children dwelling in households with livestock or other animals (Yes = 31.1%, No– 19.4%). Furthermore, parasitaemia level decreased as housing quality and wealth quintile increased.

**Percentage changes in LLIN utilisation and parasitaemia level between 2015 and 2018.** The utilisation of LLIN among the study States was 49.6% in 2015 and 53.4% in 2018. Four States: Kano, Adamawa, Delta, Katsina and Taraba recorded significant changes in LLIN utilisation between 2015 and 2018 (Table 4a).

The overall prevalence of parasitaemia in 2015 was 29.0%; 26.2% among states with increased parasitaemia and 30.4% among states with reduced levels compared with 26.4%, 30.6%, and 24.2% respectively in 2018 (Table 4b). The prevalence ratio of parasitaemia in DHS, 2018 versus MIS 2015 is shown in Fig 3. The prevalence ratio was statistically significant in Taraba and Adamawa States with reduced parasitaemia levels.

## Factors associated with utilisation of LLIN in under-five children

Unadjusted models revealed several factors related to the utilisation of LLIN in under-five children (Panel 1, Table 5). Compared to children of household head, grandchildren (OR = 0.56, 95%CI: 0.42–0.75) and other relatives (OR = 0.56, 95%CI: 0.38–0.82) were less likely to sleep under LLIN. In contrast, stunted (OR = 1.16, 95%CI: 1.00–1.34) and wasted (OR = 1.40, 95% CI: 1.05–1.88) children were more likely of sleeping under LLIN compared to normal children. Also, children whose mothers were aged 35–49 years were less likely to use LLIN than those with mothers aged <25 years. Surprisingly, children whose mothers listened to radio/television [at least once a week (OR = 0.79, 95%CI: 0.65–0.96)) or less than a week (OR = 0.75, 95% CI:0.61–0.93) were less likely to use LLIN compared with those who did not. Results for some

**Table 4.** a. LLIN utilization by survey years and percentage changes between 2015 and 2018 in 13 Nigeria States. b. Parasitaemia prevalence and percentage changes between 2015 and 2018 in 13 Nigeria States.

| Pattern of change in parasitaemia prevalence | State | MIS, 2015 | | DHS, 2018 | | % Change | p-value** | MIS, 2015 | | DHS, 2018 | | % Change | p-value** |
|---|---|---|---|---|---|---|---|---|---|---|---|---|---|
| | | n | % | n | % | | | n | Prevalence | n | Prevalence | | |
| Increase | Gombe | 235 | 40.50 | 379 | 35.24 | -13.0 | 0.190 | 176 | 28.8 | 233 | 30.1 | 4.7 | 0.768 |
| | Jigawa | 300 | 76.70 | 399 | 80.92 | 5.5 | 0.176 | 254 | 28.0 | 269 | 34.5 | 23.2 | 0.109 |
| | Kano | 231 | 58.39 | 559 | 69.95 | 19.8 | 0.002 | 171 | 26.9 | 341 | 32.8 | 22.2 | 0.168 |
| | Ogun | 160 | 17.93 | 277 | 21.61 | 20.5 | 0.357 | 118 | 14.4 | 176 | 20.7 | 44.4 | 0.163 |
| Reduction | Adamawa | 255 | 32.24 | 291 | 49.15 | 52.5 | 0.000 | 208 | 35.2 | 192 | 17.0 | -51.7 | 0.000 |
| | Delta | 151 | 20.24 | 197 | 32.51 | 60.6 | 0.010 | 132 | 20.5 | 122 | 14.8 | -27.8 | 0.235 |
| | Kaduna | 233 | 66.62 | 405 | 64.34 | -3.4 | 0.560 | 200 | 36.8 | 264 | 33.9 | -7.9 | 0.517 |
| | Katsina | 307 | 57.65 | 456 | 71.52 | 24.1 | 0.000 | 234 | 27.4 | 272 | 24.0 | -12.5 | 0.378 |
| | Kwara | 166 | 20.26 | 237 | 22.41 | 10.6 | 0.605 | 115 | 25.8 | 161 | 17.5 | -32.1 | 0.095 |
| | Niger | 205 | 38.07 | 424 | 38.50 | 1.1 | 0.918 | 176 | 33.6 | 255 | 31.4 | -6.6 | 0.626 |
| | Osun | 120 | 23.01 | 246 | 29.74 | 29.3 | 0.175 | 86 | 34.2 | 151 | 26.4 | -22.7 | 0.205 |
| | Taraba | 246 | 33.66 | 302 | 24.10 | -28.4 | 0.013 | 212 | 43.3 | 189 | 19.9 | -54.1 | 0.000 |
| | Yobe | 282 | 56.12 | 359 | 58.68 | 4.5 | 0.516 | 246 | 18.8 | 219 | 12.7 | -32.3 | 0.074 |

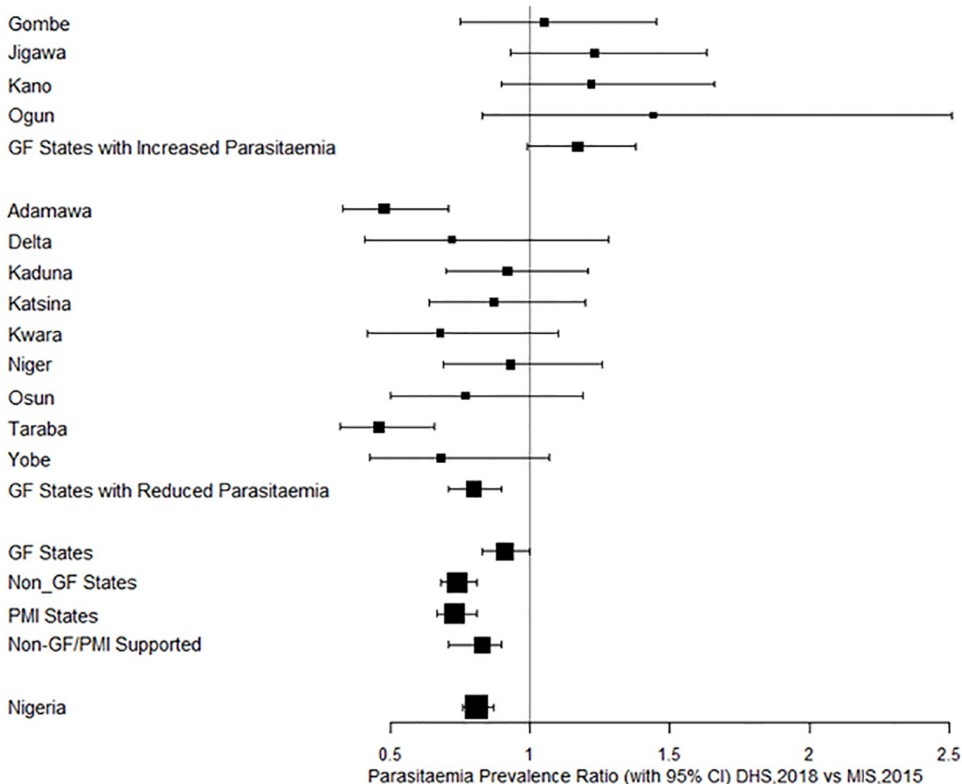

**Fig 3. Forest Plot of Prevalence Ratio with 95% CI in 2018 vs 2015.**

perceptions/opinions about malaria showed that the following opinions were associated with LLIN use: preventative medicine keeps baby healthy (OR = 1.31, 95%CI: 1.06–1.63)), malaria can lead to death (OR = 1.31, 95%CI: 1.12–1.52), only weak children can die from malaria (OR = 1.67, 95%CI: 1.41–1.98). Female household headship was negatively related to LLIN use in under-fives. Mothers who obtained LLIN from antenatal/immunisation clinic were more likely to use it for their children than women who obtain net from other sources (OR = 4.33, 95%CI: 2.06–9.11). Other factors positively associated with LLIN use include rural residence (OR = 1.51, CI: 1.09–2.08); and high community disadvantage (OR = 3.53, CI: 2.45–5.09).

To identify independent predictors, multivariable models were fitted and presented in Table 5. Panel 2 showed results of all the 13 States. Grandchildren (AOR = 5.35, 95%CI: 1.09–26.19) were more likely to use LLIN while other relatives (AOR = 0.33, 95%CI: 0.11–0.94) were less likely compared to direct children of household heads. Furthermore, LLIN use was higher in children whose mothers opined that only weak children could die from malaria (AOR = 1.83, 95%CI: 1.10–3.10). LLIN use was less likely where the household head is a female (AOR = 0.51, 95%CI: 0.27–0.99); the quality of housing materials was poor (AOR = 0.25, 95% CI: 0.09–0.72) and the mothers listened to radio/television less than once a week (OR = 0.46, 95%CI:0.26–0.81). In contrast, utilisation of LLIN is more likely among children whose mothers obtained net from ANC or immunisation clinics (AOR = 5.30, 95%CI: 2.32–12.14) and campaigns (AOR = 1.77, 95%CI: 1.03–3.04). High community disadvantage was a significant predictor of LLIN utilisation (AOR = 10.91, CI: 3.12–38.06).

Replication of the multivariable model for states with increased parasitaemia between 2015 and 2018 (Table 5, Panel 3) revealed that other household head relatives were less likely to use

**Table 5. Factors associated with LLIN utilisation among under-5 children in 13 selected States, Nigeria (NDHS 2018).**

| Individual child/maternal characteristics | Unadjusted OR (95% CI) | p value | All States | | States with Malaria Increase | | States with Malaria reduction | |
|---|---|---|---|---|---|---|---|---|
| | | | Adjusted OR (95% CI) | p value | Adjusted OR (95% CI) | p value | Adjusted OR (95% CI) | p value |
| *Child's age (Months)* | | | | | | | | |
| 0–11 | 1.00 | | | | | | | |
| 12–23 | 0.99 (0.80–1.23) | 0.96 | | | | | | |
| 24–59 | 0.85 (0.72–1.02) | 0.078 | | | | | | |
| *Child's sex* | | | | | | | | |
| Male | 1.00 | | | | | | | |
| Female | 0.97 (0.85–1.12) | 0.689 | | | | | | |
| *Relationship to head of household* | | | | | | | | |
| Child | 1.00 | | | | | | | |
| Grandchild | 0.56 (0.42–0.75)* | <0.001 | 5.35 (1.09–26.19)* | 0.039 | 4.14 (0.47–36.62) | 0.202 | 4.20 (0.24–72.74) | 0.324 |
| Other relative | 0.56 (0.38–0.82)* | 0.003 | 0.33 (0.11–0.94)* | 0.038 | 0.22 (0.07–0.76)* | 0.016 | 5.22 (0.25–107.61) | 0.284 |
| *Fever in past 2 weeks (Yes vs No)* | 1.09 (0.92–1.28) | 0.328 | | | | | | |
| *Stunting (Yes vs No)* | 1.16 (1.00–1.34) | 0.055 | 0.75 (0.50–1.13) | 0.166 | 0.82 (0.51–1.31) | 0.406 | 0.64 (0.22–1.83) | 0.408 |
| *Wasting (Yes vs No)* | 1.40 (1.05–1.88)* | 0.022 | 1.25 (0.57–2.72) | 0.582 | 0.93 (0.38–2.27) | 0.881 | 4.18 (0.34–51.65) | 0.264 |
| *Age of mother (Years)* | | | | | | | | |
| < 25 | 1.00 | | | | | | | |
| 25–34 | 0.77 (0.64–0.93)* | 0.006 | 0.92 (0.53–1.59) | 0.753 | 0.78 (0.42–1.45) | 0.429 | 3.60 (0.98–13.27) | 0.054 |
| 35–49 | 0.68 (0.55–0.84)* | <0.001 | 0.70 (0.38–1.30) | 0.259 | 0.65 (0.33–1.29) | 0.218 | 3.39 (0.81–14.11) | 0.094 |
| *Mother's education* | | | | | | | | |
| None | 1.00 | | | | | | | |
| Primary | 1.19 (0.95–1.50) | 0.138 | | | | | | |
| Secondary/higher | 0.97 (0.78–1.21) | 0.811 | | | | | | |
| *Mother's occupation* | | | | | | | | |
| Not working | 1.00 | | | | | | | |
| White collar job | 1.09 (0.81–1.47) | 0.559 | | | | | | |
| Sales | 0.86 (0.73–1.02) | 0.09 | | | | | | |
| Agric/manual work | 0.84 (0.67–1.06) | 0.144 | | | | | | |
| *Exposure to radio/television* | | | | | | | | |
| Not at all | 1.00 | | | | | | | |
| less than once a week | 0.75 (0.61–0.93)* | 0.007 | 0.46 (0.26–0.81)** | 0.007 | 0.65 (0.32–1.28) | 0.212 | 0.13 (0.03–0.54)* | 0.005 |
| at least once a week | 0.79 (0.65–0.96)* | 0.02 | 0.61 (0.34–1.10) | 0.098 | 0.84 (0.43–1.67) | 0.621 | 0.58 (0.13–2.57) | 0.475 |
| *Health Insurance* | | | | | | | | |
| Yes vs No | 1.17 (0.65–2.10) | 0.599 | | | | | | |
| **Perceptions/opinions about malaria** | | | | | | | | |
| Preventative medicine keeps baby healthy | 1.31 (1.06–1.63)* | 0.014 | 1.71 (0.60–4.90) | 0.318 | | | 19.92 (4.15–95.67)* | <0.001 |
| Malaria can be fully cured by medicine | 1.17 (0.97–1.41) | 0.095 | | | | | | |
| Malaria can lead to death | 1.31 (1.12–1.52)* | 0.001 | 1.55 (0.95–2.63) | 0.089 | 2.52 (1.32–4.83)* | 0.005 | 0.63 (0.19–2.06) | 0.448 |
| No worry about malaria due to easy treatment | 1.01 (0.86–1.18) | 0.906 | | | | | | |
| I know people sick with malaria | 1.08 (0.93–1.26) | 0.307 | | | | | | |
| Only weak children can die from malaria | 1.67 (1.41–1.98)* | <0.001 | 1.83 (1.10–3.10)* | 0.023 | 0.99 (0.54–1.83) | 0.9995 | 15.64 (2.88–85.05)* | 0.001 |
| **Household characteristics** | | | | | | | | |
| *Sex of household head* | | | | | | | | |

(*Continued*)

**Table 5.** (Continued)

| Individual child/maternal characteristics | Unadjusted OR (95% CI) | p value | All States | | States with Malaria Increase | | States with Malaria reduction | |
|---|---|---|---|---|---|---|---|---|
| | | | Adjusted OR (95% CI) | p value | Adjusted OR (95% CI) | p value | Adjusted OR (95% CI) | p value |
| Male | 1.00 | | | | | | | |
| Female | 0.69 (0.53–0.91)* | 0.008 | 0.51 (0.27–0.99)* | 0.046 | 0.54 (0.25–1.18) | 0.122 | 0.69 (0.12–3.85) | 0.669 |
| *Quality of housing material* | | | | | | | | |
| Good | | | | | | | | |
| Average | 1.58 (1.30–1.92)* | <0.001 | 0.53 (0.29–0.98)* | 0.041 | 0.27 (0.12–0.63)* | 0.002 | 2.13 (0.57–7.98) | 0.264 |
| Poor | 1.28 (0.94–1.75) | 0.122 | 0.25 (0.09–0.72)* | 0.009 | 0.12 (0.03–0.48)* | 0.003 | 0.21 (0.02–2.19) | 0.192 |
| *Wealth index* | | | | | | | | |
| Poor | 1.23 (0.94–1.61) | 0.13 | | | | | | |
| Middle | 1.04 (0.82–1.31) | 0.767 | | | | | | |
| Rich | 1.00 | | | | | | | |
| *Obtained net from campaigns* | | | | | | | | |
| No | 1.00 | | | | | | | |
| Yes, campaign | 1.39 (0.85–2.27) | 0.053 | 1.77 (1.03–3.04)* | 0.038 | 1.21 (0.62–2.32) | 0.577 | 7.25 (2.02–26.05)* | 0.002 |
| ANC / Immunization clinic | 4.33 (2.06–9.11)* | <0.001 | 5.30 (2.32–12.14)* | <0.001 | 4.32 (1.58–11.85)* | 0.004 | 7.94 (1.29–48.81)* | 0.025 |
| *Number of months ago net was obtained* | | | | | | | | |
| <=2 vs >2 | 0.86 (0.45–1.63) | 0.644 | | | | | | |
| *Number of under5 in the household* | | | | | | | | |
| 1 | 1.00 | | | | | | | |
| 2 | 1.11 (0.92–1.34) | 0.285 | 1.05 (0.62–1.76) | 0.865 | 1.13 (0.60–2.13) | 0.695 | 1.31 (0.44–3.89) | 0.628 |
| >=3 | 0.77 (0.62–0.94)* | 0.012 | 0.68 0.36–1.28) | 0.229 | 0.71 (0.34–1.49) | 0.37 | 0.73 (0.18–3.07) | 0.671 |
| *No. of household member per net* | | | | | | | | |
| <=2 | 1.80 (1.48–2.19)* | <0.001 | 0.88 (0.58–1.33) | 0.548 | 0.97 (0.59–1.58) | 0.887 | 0.97 (0.35–2.72) | 0.957 |
| >2 | 1.00 | | 1 | | 1 | | | |
| *Ownership of radio/tv in household* | | | | | | | | |
| Yes vs No | 1.07 (0.89–1.28) | 0.502 | | | | | | |
| **Community (cluster) characteristics** | | | | | | | | |
| *Location of residence* | | | | | | | | |
| Rural vs Urban | 1.51 (1.09–2.08)* | 0.012 | 0.72 (0.27–1.94) | 0.526 | 0.31 (0.08–1.25) | 0.101 | 3.28 (0.59–18.34) | 0.176 |
| *Community poverty* | | | | | | | | |
| Low | 1.00 | | | | | | | |
| High | 0.87 (0.64–1.19) | 0.382 | | | | | | |
| *Community illiteracy* | | | | | | | | |
| Low | 1.00 | | | | | | | |
| High | 1.21 (0.89–1.65) | 0.227 | | | | | | |
| *Community disadvantage* | | | | | | | | |
| Low | 1.00 | | | | | | | |
| Medium | 2.08 (1.42–3.06)* | <0.001 | 5.18 (1.67–16.06)* | 0.004 | 11.37 (2.06–62.79) | 0.005 | 3.87 (0.58–25.73) | 0.162 |
| High | 3.53 (2.45–5.09)* | <0.001 | 10.91 (3.12–38.06)* | <0.001 | 51.67 (8.03–332.52)* | <0.001 | 9.61 (0.87–106.57) | 0.065 |

* p<0.05.

LLIN compared to biological children (AOR = 0.22, CI: 0.07–0.76). On perceptions, the use of LLIN was more likely in children whose mothers opined that malaria could lead to death (AOR = 2.52, CI: 1.32–4.83). LLIN use was less common among children from households with poor quality building materials (AOR = 0.12, CI: 0.03–0.48). The likelihood of LLIN use for under-five was significantly higher when mothers obtained nets from ANC/immunisation clinics.

In the nine states with the reduction in parasitaemia between 2015 and 2018, significant predictors of LLIN use included some opinions about malaria viz: preventative medicine keeps baby healthy, and only weak children can die from malaria. Further, LLIN use was higher in children whose mothers obtained net from campaigns (AOR = 7.25, CI: 2.02–26.05) and ANC/immunisation clinic (AOR = 7.94, CI: 1.29–48.81). Children whose mothers listened to radio/television less than once a week were less likely to use LLIN (OR = 0.13, (0.03–0.54) compared with those who did not.

### Factors associated with parasitaemia in under-five children

Results from random effect logit models for factors associated with parasitaemia are presented in Table 6. The unadjusted models showed several variables attained statistical significance. Of these, child characteristics included age of the child, the relationship of the child to head of household, fever in the past two weeks, and stunting.

In the adjusted model for the 13 States (Table 6, Panel 2), variables found to be predictors of parasitaemia include child age, of which those aged 24–59 months were almost two times as likely to have parasitaemia compared to children aged 0–11 months (AOR = 1.78, CI: 1.28–2.48). Similarly, children in whom fever was reported were more likely to have parasitaemia (AOR = 1.31, CI: 1.06–1.63). Children of women with no formal education were also about two times as likely to have parasitaemia compared to those whose mothers attained secondary/higher education. The same pattern was observed for children from poor households compared to the rich (AOR = 2.06, CI: 1.24–3.42). The odds of parasitaemia were found to be 98% higher among rural children relative to their urban counterparts (AOR = 1.98, CI: 1.37–2.87). Similarly, children who live in a community with a high level of illiteracy had higher odds of parasitaemia (AOR = 10.91, CI: 3.12–38.06).

The adjusted model for the four States where parasitaemia prevalence increased between 2015 and 2018 is summarized in panel 3 of Table 6. In these four states, children aged 24–59 months were more likely of parasitaemia relative to those aged 0–11 months (AOR = 2.37, CI: 1.36–4.13). Fever in the past two weeks remained a significant predictor. Furthermore, parasitaemia in under-five children was associated with a lack of formal education among mothers (AOR = 1.89, CI: 1.03–3.45). Medium (AOR = 2.84, CI: 1.07–7.54) and high (AOR = 3.93, CI: 1.09–14.21) levels of community disadvantage were parasitaemia predictors.

Adjusted models for the nine States which had a reduction in parasitaemia prevalence between 2015 and 2018 are summarised in panel 4 of Table 6. Grandchildren were twice as likely to be parasitaemia positive compared to children of household heads (AOR = 2.03, CI: 1.14–3.59). Other significant variables included lack of formal education by mother (AOR = 1.99, CI: 1.26–3.13), poor household wealth index (AOR = 2.62, CI: 1.37–5.02), rural versus urban residence (AOR = 1.93, CI: 1.19–3.15), and high community illiteracy (AOR = 1.86, CI: 1.24–2.79). Children in settings with high community disadvantage were found to be less likely of having parasitaemia (AOR = 0.40, CI: 0.19–0.82).

## Discussion

In this study, we explored changes in parasitaemia prevalence and investigated factors associated with parasitaemia and LLIN use in 13 States with high malaria burden in Nigeria. In

**Table 6. Factors associated with parasitaemia among under-5 Children in 13 selected States of Nigeria (NDHS 2018).**

| Individual child/maternal factors | Unadjusted OR (95% CI) | p values | All States | | States with Malaria Increase | | States with Malaria reduction | |
|---|---|---|---|---|---|---|---|---|
| | | | Adjusted OR (95% CI) | p values | Adjusted OR (95% CI) | p value | Adjusted OR (95% CI) | p value |
| *Child's age (Months)* | | | | | | | | |
| 0–11 | 1.00 | | 1.00 | | 1.00 | | 1.00 | |
| 12–23 | 1.08 (0.76–1.54) | 0.677 | 0.94 (0.65–1.36) | 0.735 | 1.03 (0.55–1.94) | 0.916 | 0.87 (0.54–1.39) | 0.561 |
| 24–59 | 1.92 (1.40–2.64)* | <0.001 | 1.78 (1.28–2.48)* | 0.001 | 2.37 (1.36–4.13)* | 0.002 | 1.47 (0.97–2.22) | 0.073 |
| *Child's sex* | | | | | | | | |
| Male | | | | | | | | |
| Female | 0.86 (0.72–1.04) | 0.114 | | | | | | |
| *Relationship to head of household* | | | | | | | | |
| Child | 1.00 | | 1.00 | | 1.00 | | 1.00 | |
| Grandchild | 1.42 (1.02–2.00)* | 0.04 | 1.51 (0.94–2.43) | 0.088 | 0.87 (0.35–2.13) | 0.756 | 2.03 (1.14–3.59)* | 0.016 |
| Other relative | 1.43 (0.88–2.34) | 0.153 | 1.72 (0.87–3.39) | 0.117 | 2.15 (0.88–5.28) | 0.094 | 0.87 (0.27–2.80) | 0.821 |
| *Anaemia* | | | | | | | | |
| Severe | 15.37 (9.01–26.20)* | <0.001 | | | | | | |
| Moderate | 4.92 (3.84–6.31)* | <0.001 | | | | | | |
| Mild | 1.98 (1.50–2.62)* | <0.001 | | | | | | |
| None | | | | | | | | |
| *Use of LLIN (Yes vs No)* | 0.94 (0.77–1.14) | 0.514 | 0.84 (0.67–1.03) | 0.107 | 0.76 (0.53–1.09) | 0.136 | 0.86 (0.65–1.13) | 0.284 |
| *Fever in past 2 weeks (Yes vs No)* | 1.33 (1.08–1.65)* | 0.008 | 1.31 (1.06–1.63)* | 0.013 | 1.74 (1.22–2.49)* | 0.002 | 1.13 (0.86–1.49) | 0.384 |
| *Medical treatment of fever* | 0.69 (0.47–1.00) | 0.047 | | | | | | |
| *Stunting (Yes vs No)* | 1.28 (1.06–1.55)* | 0.01 | 1.12 (0.90–1.38) | 0.303 | 1.26 (0.89–1.79) | 0.187 | 1.05 (0.80–1.38) | 0.736 |
| *Wasting (Yes vs No)* | 1.17 (0.81–1.69) | 0.39 | | | | | | |
| *Age of mother (Years)* | | | | | | | | |
| < 25 | 1.00 | | | | | | | |
| 25–34 | 1.13 (0.88–1.45) | 0.347 | | | | | | |
| 35–49 | 1.19 (0.90–1.58) | 0.217 | | | | | | |
| *Mother's education* | | | | | | | | |
| None | 3.09 (2.33–4.11) | <0.001 | 1.89 (1.32–2.70)* | <0.001 | 1.89 (1.03–3.45)* | 0.039 | 1.99 (1.26–3.13)* | 0.003 |
| Primary | 2.11 (1.49–2.99) | <0.001 | 1.54 (1.06–2.22)* | 0.022 | 1.77 (0.94–3.32) | 0.078 | 1.46 (0.92–2.32) | 0.111 |
| Secondary/higher | 1.00 | | 1.00 | | 1.00 | | 1.00 | |
| *Mother's occupation* | | | | | | | | |
| Not working | 1.00 | | | | | | | |
| White collar job | 0.70 (0.46–1.06) | 0.092 | | | | | | |
| Sales | 1.03 (0.83–1.2)) | 0.784 | | | | | | |
| Agric/manual work | 0.79 (0.59–1.07) | 0.127 | | | | | | |
| *Mother's involvement in decision-making* | | | | | | | | |
| Poor | | | | | | | | |
| Average | 1.32 (0.99–1.75) | 0.06 | 1.53 (1.13–2.07)* | 0.005 | 1.50 (0.96–2.33) | 0.075 | 1.48 (0.97–2.26) | 0.066 |
| Good | 0.89 (0.70–1.14) | 0.376 | 1.21 (0.92–1.58) | 0.172 | 1.53 (0.93–2.54) | 0.095 | 1.10 (0.79–1.53) | 0.571 |
| *Exposure to radio/television* | | | | | | | | |
| Not at all | 1.00 | | 1.00 | | 1.00 | | 1.00 | |
| less than once a week | 0.80 (0.61–1.04) | 0.1 | 1.04 (0.79–1.37) | 0.786 | 1.00 (0.63–1.56) | 0.984 | 1.01 (0.69–1.47) | 0.953 |
| at least once a week | 0.67 (0.52–0.85)* | 0.001 | 1.18 (0.89–1.55) | 0.246 | 1.01 (0.66–1.54) | 0.969 | 1.30 (0.89–1.90) | 0.169 |
| *Health Insurance* | | | | | | | | |
| Yes vs No | 0.19 (0.05–0.68)* | 0.011 | 0.47 (0.13–1.71) | 0.255 | 0.75 (0.15–3.88) | 0.734 | 0.24 (0.03–2.05) | 0.191 |

*(Continued)*

**Table 6.** (Continued)

| Individual child/maternal factors | Unadjusted OR (95% CI) | p values | All States | | States with Malaria Increase | | States with Malaria reduction | |
|---|---|---|---|---|---|---|---|---|
| | | | Adjusted OR (95% CI) | p values | Adjusted OR (95% CI) | p value | Adjusted OR (95% CI) | p value |
| **Household characteristics** | | | | | | | | |
| *Age of household head (Year)* | | | | | | | | |
| <25 | 1.00 | | | | | | | |
| 25–34 | 0.80 (0.43–1.47) | 0.473 | | | | | | |
| 35–49 | 0.92 (0.50–1.67) | 0.774 | | | | | | |
| >=50 | 1.37 (0.75–2.52) | 0.31 | | | | | | |
| *Sex of household head* | | | | | | | | |
| Male | 1.00 | | | | | | | |
| Female | 0.89 (0.63–1.26) | 0.51 | | | | | | |
| *Quality of housing material* | | | | | | | | |
| Good | 1.00 | | | | | | | |
| Average | 1.67 (1.31–2.11)* | <0.001 | 0.98 (0.73–1.34) | 0.922 | 0.78 (0.44–1.37) | 0.382 | 1.12 (0.77–1.63) | 0.56 |
| Poor | 2.04 (1.45–2.89)* | <0.001 | 0.71 (0.45–1.13 | | 0.76 (0.35–1.66) | 0.495 | 0.65 (0.36–1.17) | 0.15 |
| *Wealth index* | | | | | | | | |
| Poor | 4.01 (2.93–5.49)* | <0.001 | 2.06 (1.24–3.42)* | 0.005 | 1.31 (0.55–3.09) | 0.542 | 2.62 (1.37–5.02)* | 0.004 |
| Middle | 2.43 (1.78–3.31)* | <0.001 | 1.48 (0.99–2.22) | 0.054 | 1.31 (0.65–2.62) | 0.446 | 1.66 (1.00–2.77) | 0.05 |
| Rich | 1.00 | | 1.00 | | 1.00 | | 1.00 | |
| *Drinking water source* | | | | | | | | |
| Not improved vs Improved | 1.25 (1.00–1.56) | 0.052 | 0.93 (0.73–1.19) | 0.585 | 0.85 (0.57–1.26) | 0.423 | 1.05 (0.76–1.46) | 0.749 |
| *Toilet type* | | | | | | | | |
| Not improved vs improved | 1.61 (1.30–1.99)* | <0.001 | 1.11 (0.86–1.43) | 0.42 | 1.09 (0.72–1.67) | 0.678 | 1.19 (0.86–1.65) | 0.283 |
| Availability of livestock around the house | | | | | | | | |
| Yes | 1.53 (1.23–1.89)* | <0.001 | 1.09 (0.85–1.40) | 0.506 | 1.02 (0.62–1.66) | 0.95 | 1.15 (0.86–1.55) | 0.347 |
| No | 1.00 | | 1.00 | | 1.00 | | 1.00 | |
| Ownership of radio/tv in household | | | | | | | | |
| Yes vs No | 0.80 (0.63–1.01) | | | | | | | |
| Community (Cluster) characteristics | | | | | | | | |
| Altitude | | | | | | | | |
| Low land (< 200m) | 1.00 | | 1.00 | | 1.00 | | 1.00 | |
| Plain land (200-600m above sea level) | 1.57 (1.11–2.23)* | 0.011 | 1.16 (0.79–1.71) | 0.443 | 0.39 (0.13–1.13) | 0.084 | 1.30 (0.82–2.07) | 0.27 |
| High land (> 600m) | 1.28 (0.77–2.13) | 0.347 | 1.13 (0.67–1.90) | 0.647 | 0.74 (0.13–4.12) | 0.727 | 1.38 (0.77–2.48) | 0.285 |
| Location of residence | | | | | | | | |
| Rural vs Urban | 2.94 (2.20–3.92)* | <0.001 | 1.98 (1.37–2.87)* | <0.001 | 1.68 (0.92–3.07) | 0.092 | 1.93 (1.19–3.15)* | 0.008 |
| Community poverty | | | | | | | | |
| Low | 1.00 | | 1.00 | | 1.00 | | 1.00 | |
| High | 1.37 (1.04–1.82)* | 0.026 | 1.18 (0.86–1.62) | 0.311 | 0.93 (0.53–1.63) | 0.798 | 1.13 (0.73–1.74) | 0.589 |
| Community illiteracy | | | | | | | | |
| Low | 1.00 | | 1.00 | | 1.00 | | 1.00 | |
| High | 2.76 (2.11–3.62)* | <0.001 | 1.97 (1.43–2.73)* | <0.001 | 1.71 (0.89–3.31) | 0.108 | 1.86 (1.24–2.79)* | 0.003 |
| Community disadvantage | | | | | | | | |
| Low | 1.00 | | 1.00 | | 1.00 | | 1.00 | |

(*Continued*)

**Table 6.** (Continued)

| Individual child/maternal factors | Unadjusted OR (95% CI) | p values | All States | | States with Malaria Increase | | States with Malaria reduction | |
|---|---|---|---|---|---|---|---|---|
| | | | Adjusted OR (95% CI) | p values | Adjusted OR (95% CI) | p value | Adjusted OR (95% CI) | p value |
| Medium | 1.91 (1.32–2.77)* | 0.001 | 0.83 (0.53–1.30) | 0.416 | 2.84 (1.07–7.54)* | 0.036 | 0.61 (0.35–1.07) | 0.085 |
| High | 3.26 (2.30–4.62)* | <0.001 | 0.63 (0.36–1.11) | 0.107 | 3.93 (1.09–14.21)* | 0.037 | 0.40 (0.19–0.82)* | 0.013 |

* p<0.05.

addition, determinants of parasitaemia and LLIN utilisation were investigated in States with increase and those with reduction in parasitaemia between 2015 and 2018. This study showed some socioeconomic differences between the states with reduced parasitaemia and those with increased parasitaemia. For example, those with increased parasitaemia has the highest proportion of under-five in the poor wealth tertile while for states with reduction, the highest proportion was in the middle wealth tertile. A higher proportion of those with totally improved housing quality was found among states with reduced parasitaemia. These inherent differences may be the main underlying factor responsible for the delayed progress in malaria outcomes in the four states.

Among states where parasitaemia levels increased between 2015 and 2018, factors associated with utilisation of LLIN were mostly socioeconomic and behavioural, and these corroborate findings from previous studies [7,14]. The factors include source of net (obtaining net from ANC/immunisation clinic), high community disadvantage and perceived severity of malaria which, positively influenced LLIN use while poor-quality housing [8,15] and being a non-biological child of household head were negatively associated with LLIN use. Those who obtained nets from immunisation clinics had higher odds of use compared to those who obtained from other sources. A plausible reason for this could be that distribution at ANC/immunisation clinic may have been accompanied by health education which encouraged usage.

Factors that influence malaria prevalence are complex, ranging from micro-level peculiarities of individuals to macro-level factors on national, international, and global scales [16]. In this study, the factors found to be predictors of parasitaemia include the age of the child (24–59 months), having fever in the past two weeks, lack of formal education among mothers, medium and high level of community disadvantage.

At the individual level, we found older age to be significantly associated with higher odds of malaria infection in states that recorded an increase in parasitaemia during the period of investigation. This finding is in line with those of previous studies [8,17,18]. It has been shown scientifically that children in areas of high malaria transmission intensity develop age-related immunity [18,19]. First, they are protected from malaria by acquired immunity from their mothers, but this acquired immunity gradually fades as the children grow [19] and thereafter, the continuous exposure to infective mosquito bites lead to the development of immunity [20]. This explains the asymptomatic state that older children will more likely have malaria parasites. This explanation, coupled with the likelihood of a more proactive attitude towards malaria prevention among caregivers for the youngest children and the focus of National malaria programs on the younger children for a long time, may explain why the older ones tend to be more susceptible to malaria infection.

Fever in the past two weeks was a significant driver of parasitaemia among states with increase in parasitaemia level. This finding reflects the malaria infection intensity. At

population level, fever is an important indicator of levels of malaria transmission and malaria risks in the communities [21]. With the high level of transmission, a high incidence of fever is expected as those infected are likely to develop symptoms. Prevalence of fever and malaria infection directly impacts malaria case management and the use of antimalarials. In this study, we found that those who reported seeking medical treatment for fever were less likely to have parasitaemia, although not statistically significant. However, this is in line with the new guideline of testing to confirm malaria before treatment [22].

Children of educated mothers had lower odds of malaria parasitaemia. Maternal education is a key determinant of the health of under-five children. Education affects the perception of malaria preventive measures, including the acceptability and practice of malaria control interventions [23]. A putative causal relationship has been reported for the impact of a mother's level of education on under-five malaria parasitaemia [24]. Mothers who had attained higher education are more likely to have greater exposures to means and methods of living a healthier life and specifically to prevent and treat malaria [25]. The significance of higher education attainment of a child's mother and better wealth status of the household in which a child dwell in reducing the risk of a child's ill health, including malaria, is well documented in the literature [26,27].

The role of socio-economic development in malaria transmission cannot be overemphasised. Sachs and Malaney, 2002 highlighted a "striking and unmistakable correlation between malaria and poverty" at the national level [28]. Moreover, improved socioeconomic circumstances have been listed as one of the major driving forces for success among the 34 countries that have made progress in malaria elimination between 2000 and 2015 [29]. Because wealth impacts other indices like education, housing, household nutrition, areas of residence and health-seeking behaviour, it is arguably a major determinant of malaria in under-fives [23].

At the community level, living in the most disadvantaged communities predisposes an under-five to parasitaemia. Community disadvantage is a composite of rural residency, no formal education in mother, and poor household wealth quintile. All these factors have been demonstrated to be individually associated with parasitaemia levels. Dickinson et al suggested three pathways through which individual and household socioeconomic status (SES) are related to malaria and subsequent health status [30]. The first pathway was that of SES affecting access to malaria prevention. The second was that of SES being a fundamental cause of malaria, through poor housing quality and increased psychological stress, which is linked to lower immunity and subsequent susceptibility to infection. The third pathway proposed was that of SES affecting "access to accurate diagnosis and effective malaria treatment.

The inability of this study to demonstrate a significant association between the use of LLIN and parasitaemia does not indicate that the former is unimportant. This is because reported use may not be a true representation of actual use [31]. However, the results also suggest that LLIN utilisation may not be at an optimal level sufficient to impact parasitaemia prevalence positively. The LLINs are typically protective indoors, but factors such as the inconvenience experienced in setting up [32], entering and exiting the bed nets; discomfort associated with heat and low malaria risk perception have been shown to contribute to a lack of consistency in its use [33]. In addition, staying late outdoor and sleeping outside the bed prior to retiring to bed favour outdoor biting and may limit the protective capacity of mosquito nets. Individuals residing in rural areas are at particular risk of exposure to outdoor biting, as factors such as the absence of electricity for indoor lighting and the discomfort of indoor heat may force individuals to stay out longer at dusk or even sleep outside [32].

### Strengths and limitations

This study was able to identify individual, household, and community-level predictors of malaria parasitaemia among under-fives in selected states of Nigeria. The findings thus provide information that can be useful in the planning and designing appropriate and targeted interventions for malaria elimination. Furthermore, the surveys were conducted about the same period, thus reducing confounding due to seasonality. Importantly, microscopy was conducted under quality-controlled conditions in the same accredited laboratory.

Given that the data resulted from cross-sectional designs, a causal relationship between explanatory variables and parasitaemia cannot be assumed. Another limitation was that other data on climatic and environmental conditions such as rainfall, humidity, and temperature as well as the status of insecticide resistance in the states were not available for inclusion in the analyses.

### Conclusion

This study showed that LLIN use was poor. Adoption of interventions, especially those requiring behavioural changes, is challenging. This may be a plausible reason for the poor LLIN use. Health promotion activities and mass public health campaigns have often failed to have the desired effect in terms of reducing disease incidence and burden, simply because compliance with the message, in the form of the intended behaviour change, is harder to achieve than its precursors of raising awareness, providing knowledge and changing attitudes. This calls for serious scrutiny of the method of delivery of messages by control programmes and behaviour of the populace.

Observation of differential changes in the level of parasitaemia and LLIN use over time in the study states and the varied drivers of this change are pointers to the fact that malaria control and ultimately eradication is not an isolated effort of malaria control programmes, but part of a holistic approach of improving education and socioeconomic status of the population.

### Implications of findings for policy and programmes

The national malaria operations research agenda should consider intervention studies to address gaps identified in this analysis. For example, an innovative approach to encouraging LLIN use and assessing its effectiveness can be developed. In addition, there can be a collaboration between Ministries of Environment and Housing to develop strategiesto encourage building "quality" and not necessarily expensive houses.

The findings of this analysis highlighted the complexity of malaria control. The involvement of the people, community and the Government is paramount. Therefore, there is a need for synergy of support from development partners, funding agencies, and relevant government ministries. In choosing states to support, the development partners should target both states doing well in terms of control and those not doing so to sustain the gains of control and institute measures to address gaps in the states. The development partners should provide malaria commodities and enhance behavioural change communications to complement the supply of antimalarial, LLINs and diagnostics.

### Author Contributions

**Conceptualization:** Perpetua Uhomoibhi, IkeOluwapo O. Ajayi, Olugbenga Mokuolu, Adeniyi Fagbamigbe, Joshua O. Akinyemi.

**Data curation:** Adeniyi Fagbamigbe, Joshua O. Akinyemi.

**Formal analysis:** Adeniyi Fagbamigbe, Joshua O. Akinyemi.

**Methodology:** IkeOluwapo O. Ajayi, Olugbenga Mokuolu, Ibrahim Maikore, Adeniyi Fagbamigbe, Joshua O. Akinyemi.

**Resources:** Chukwu Okoronkwo, Ibrahim Maikore, Festus Okoh, Cyril Ademu, Issa Kawu, Jo-Angeline Kalambo, James Ssekitooleko.

**Supervision:** Perpetua Uhomoibhi, IkeOluwapo O. Ajayi, Olugbenga Mokuolu.

**Visualization:** Adeniyi Fagbamigbe.

**Writing – original draft:** IkeOluwapo O. Ajayi, Adeniyi Fagbamigbe, Joshua O. Akinyemi.

**Writing – review & editing:** Perpetua Uhomoibhi, Chukwu Okoronkwo, IkeOluwapo O. Ajayi, Olugbenga Mokuolu, Ibrahim Maikore, Adeniyi Fagbamigbe, Joshua O. Akinyemi, Festus Okoh, Cyril Ademu, Issa Kawu, Jo-Angeline Kalambo, James Ssekitooleko.

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
