## [Decision Letter · Decision Letter 0]

27 Sep 2021

PONE-D-21-27138Drivers of long-lasting insecticide-treated net utilisation and parasitaemia among under-five children in 13 States with high malaria burden in NigeriaPLOS ONE

Dear Dr. Akinyemi,

Thank you for submitting your manuscript to PLOS ONE. After careful consideration, we feel that it has merit but does not fully meet PLOS ONE’s publication criteria as it currently stands. Therefore, we invite you to submit a revised version of the manuscript that addresses the points raised during the review process. Let your revisions be in a different font colour to enable tracking

We look forward to receiving your revised manuscript.

Kind regards,

Sammy O. Sam-Wobo

Academic Editor

PLOS ONE

Journal Requirements:

Additional Editor Comments (if provided):

Authors to attend to the comments and send back. Note that your revisions must be in a different font colour to enable tracking

Reviewers' comments:

Reviewer's Responses to Questions

**Comments to the Author**

1. Is the manuscript technically sound, and do the data support the conclusions?

Reviewer #1: Yes

2. Has the statistical analysis been performed appropriately and rigorously? 

Reviewer #1: Yes

3. Have the authors made all data underlying the findings in their manuscript fully available?

Reviewer #1: No

4. Is the manuscript presented in an intelligible fashion and written in standard English?

Reviewer #1: Yes

5. Review Comments to the Author

Reviewer #1: ABSTRACT –

Methods

Line 28: Data from…………….were obtained and analyzed.

You didn’t state the role of the 2015 data set (to categorize malaria burdens in states, to compare with 2018 data for increase or reduced parasitaemia?)

Line 29 - 30: The thirteen states studied were stratified into two based on increased or reduced parasitaemia between 2015 and 2018. (Remove whether they had…).

Line 43: Last paragraph of the result….. 2.70) were more likely to have parasitaemia. NOT were more likely of parasitaemia.

INTRODUCTION

Line 70: economic loses: out of pocket payment, …….. (Add colon before listing).

Line 93: For instance, parasitaemia was reported to be more prevalent………

Line 95: ……..those from rich households were reported to have lower risks (5)

Line 95 – 96: remove the statement ‘this is a direct reflection of the fact that malaria is a poverty related disease’.

Line 103 – 105: Four of these states had increased parasitaemia between 2015 and 2018, while the remaining nine experienced a decrease (Citation is required).

Line 106: What is programmatic intervention?

Line 109 – 110: the statement there is not necessary.

METHODS

Line 113: This study involves analysis of secondary data obtained from the …..OR This is a retrospective study that utilized two national data sets. The 2015 data set on NMIS was used to categorize states as having high or low malaria burden, and also to compare with 2018 data for indicating increased or reduced paraasitaemia……..

Line 113 – 115: collapse into the subtitle ‘description of data sources’

Line 119: 49 years old

Line 122: specify the sample size ; not over 8,000 households

Line 126: 15 – 45 years

Line 126 – 130: no indication of study duration, from which month to ?

Line 132: The data sets from 2015 NMIS and 2018 NDHS were both obtained from stratified samples selected in two stages.

Line 134: write EAs in full

Line 144: men aged 15 to 59 years were not mentioned before ???????

Line 144: biomarker sample? Do you mean blood sample?

Line 148 – 152: did you select the states based on rural / urban stratification? and/or on high malaria burden? Which of the data sets provided guide for the selection or description of the study population?

Line 160: Write GF in full

Line 173: Wealth quintiles? Do you mean quartiles?

RESULTS

Line 209 – 214: you did not clearly explain Table 2.

Line 211: ………., while close to one quarter. Please specify the value.

Line 216: first column in Table 2 has no title. I suggest ‘Variables / categories’

Line 216: titles of the 2nd and 3rd panels in Table 2 are not clear. ‘state with increase’? ‘states with reduction’?

Line 246 – 252: where is the result described? There is no reference to any Table or Figure.

Line 271: …….24.2 % respectively in 2018. Remove comma.

Line 272: is the difference in prevalence between 2015 and 2018 data the same as prevalence ratio?

Line 275: where is the Figure 3?

Line 281 – 282: where is the reference group in that statement? ‘In contrast, stunted (OR=1.16, 95%CI: 1.00-1.34) and wasted (OR=1.40, 95% CI:1.05-1.88) children were more likely of sleeping under LLIN, than ………..

Your result presentation focused only on 2018 data. No Table presented comparative prevalence in 2015 and 2018 to show increase or decrease in prevalence.

DISCUSSION

Line 390 – 391: ……..global scales (Allwell-Brown, 2017). This citation format is not consistent with the one you have been using. Check other parts of this work and make the citation uniform.

Line 407: an increase in parasitaemia level over. What is OVER in the sentence?

Line 428 – 429: Recast this phrase ‘Malaria eliminating countries’

Line 438 – 439: check the punctuations in that statement.

6. PLOS authors have the option to publish the peer review history of their article (what does this mean?). If published, this will include your full peer review and any attached files.

Reviewer #1: **Yes: **Dr. C. M. Egbuche

---

## [Author Response · Author response to Decision Letter 0]

3 Nov 2021

Reviewer #1: ABSTRACT –

Methods

Line 28: Data from…………….were obtained and analyzed.

You didn’t state the role of the 2015 data set (to categorize malaria burdens in states, to compare with 2018 data for increase or reduced parasitaemia?)

RESPONSE: Thank you. We have stated the role of 2015 data.

Line 29 - 30: The thirteen states studied were stratified into two based on increased or reduced parasitaemia between 2015 and 2018. (Remove whether they had…).

RESPONSE: This has been done. Thank you

Line 43: Last paragraph of the result….. 2.70) were more likely to have parasitaemia. NOT were more likely of parasitaemia.

RESPONSE: Thank you, correction has been done as recommended

INTRODUCTION

Line 70: economic loses: out of pocket payment, …….. (Add colon before listing).

RESPONSE: thank you, colon has been added before listing

Line 93: For instance, parasitaemia was reported to be more prevalent………

RESPONSE: Thank, edits has been adopted

Line 95: ……..those from rich households were reported to have lower risks (5)

RESPONSE: Thank you, suggestion has been adopted

Line 95 – 96: remove the statement ‘this is a direct reflection of the fact that malaria is a poverty related disease’.

RESPONSE: Statement has been removed

Line 103 – 105: Four of these states had increased parasitaemia between 2015 and 2018, while the remaining nine experienced a decrease (Citation is required).

RESPONSE: Citation has been provided

Line 106: What is programmatic intervention?

RESPONSE: We have revised it for better clarity

Line 109 – 110: the statement there is not necessary.

RESPONSE: This statement was meant to clearly described the objectives of the paper. We retained it so that readers can easily link our results with the objectives.

METHODS

Line 113: This study involves analysis of secondary data obtained from the …..OR This is a retrospective study that utilized two national data sets. The 2015 data set on NMIS was used to categorize states as having high or low malaria burden, and also to compare with 2018 data for indicating increased or reduced paraasitaemia……..

RESPONSE: Thank you for these suggested revisions. We have adopted it and edited the section accordingly.

Line 113 – 115: collapse into the subtitle ‘description of data sources’

RESPONSE: This has been done

Line 119: 49 years old

RESPONSE: This correction has been made. Thank you

Line 122: specify the sample size ; not over 8,000 households

RESPONSE: the actual no of households have been stated

Line 126: 15 – 45 years

RESPONSE: Thank you for pointing out this typo. We have corrected it

Line 126 – 130: no indication of study duration, from which month to ?

RESPONSE: This has been provided

Line 132: The data sets from 2015 NMIS and 2018 NDHS were both obtained from stratified samples selected in two stages.

RESPONSE: thank you for the suggested revision which we have adopted

Line 134: write EAs in full

RESPONSE: EA has been written in full

Line 144: men aged 15 to 59 years were not mentioned before ???????

RESPONSE: Usually, DHS surveys include men. However, we have deleted the mention of men in this manuscript because we did not analyse men’s data.

Line 144: biomarker sample? Do you mean blood sample?

RESPONSE: Yes, we meant blood sample. We have replaced biomarker with “blood”

Line 148 – 152: did you select the states based on rural / urban stratification? and/or on high malaria burden? Which of the data sets provided guide for the selection or description of the study population?

RESPONSE: States were selected based on malaria burden. Both datasets provided guide for description of study population

Line 160: Write GF in full

RESPONSE: Apology for the mistake. GF has been deleted

Line 173: Wealth quintiles? Do you mean quartiles?

RESPONSE: This is wealth quintiles because it has 5 categories. Quartiles would have only 4.

RESULTS

Line 209 – 214: you did not clearly explain Table 2.

RESPONSE: Pls, note that Table 2 was further described under “utilisation of LLIN” and “prevalence of parasitaemia”

Line 211: ………., while close to one quarter. Please specify the value.

RESPONSE: The actual value has been stated

Line 216: first column in Table 2 has no title. I suggest ‘Variables / categories’

RESPONSE: Thank you for the suggestion. We have revised the column title as recommended

Line 216: titles of the 2nd and 3rd panels in Table 2 are not clear. ‘state with increase’? ‘states with reduction’?

RESPONSE: The column titles has been made clearer

Line 246 – 252: where is the result described? There is no reference to any Table or Figure.

RESPONSE: We have provided reference to Tables 4a, 4b and Figure 3

Line 271: …….24.2 % respectively in 2018. Remove comma.

RESPONSE: Thank you, Comma has been removed

Line 272: is the difference in prevalence between 2015 and 2018 data the same as prevalence ratio?

RESPONSE: No, they are not the same

Line 275: where is the Figure 3?

RESPONSE: Figures are provided as separate files but included in the manuscript for review

Line 281 – 282: where is the reference group in that statement? ‘In contrast, stunted (OR=1.16, 95%CI: 1.00-1.34) and wasted (OR=1.40, 95% CI:1.05-1.88) children were more likely of sleeping under LLIN, than ………..

RESPONSE: Sentence has been revised. Thank you

Your result presentation focused only on 2018 data. No Table presented comparative prevalence in 2015 and 2018 to show increase or decrease in prevalence.

RESPONSE: Figure 3 utilized data from both MIS 2015 and DHS 2018. Besides, we have added tables 4a and 4b which showed comparison of LLIN utilization and parasitaemia prevalence as well as percentage changes in the 13 study States between 2015 and 2018

DISCUSSION

Line 390 – 391: ……..global scales (Allwell-Brown, 2017). This citation format is not consistent with the one you have been using. Check other parts of this work and make the citation uniform.

RESPONSE: Reference style has been corrected for consistency

Line 407: an increase in parasitaemia level over. What is OVER in the sentence?

RESPONSE: We have deleted the extraneous word “Over”

Line 428 – 429: Recast this phrase ‘Malaria eliminating countries’

RESPONSE: The phrase has been recasted

Line 438 – 439: check the punctuations in that statement.

RESPONSE: Thank you the error in punctuation have been corrected

---

## [Decision Letter · Decision Letter 1]

9 Mar 2022

PONE-D-21-27138R1Drivers of long-lasting insecticide-treated net utilisation and parasitaemia among under-five children in 13 States with high malaria burden in NigeriaPLOS ONE

Dear Dr. Akinyemi,

Thank you for submitting your manuscript to PLOS ONE. After careful consideration, we feel that it has merit but does not fully meet PLOS ONE’s publication criteria as it currently stands. Therefore, we invite you to submit a revised version of the manuscript that addresses the points raised during the review process.

We look forward to receiving your revised manuscript.

Kind regards,

Lucinda Shen, MSc

Staff Editor

on behalf of 

Clement Ameh Yaro

Academic Editor 

PLOS ONE

Journal Requirements:

Additional Editor Comments:

COMMENTS TO AUTHOR(S)

The manuscript is sound but there is need for minor revision in some components as elaborated in the comments by the reviewers.

REVIEWER 1:

COMMENTS

The author has effected all the corrections pointed out.

Aside that, I noted the following minor corrections which I did not point out the first time (Please I apologize for that).

1. Add title to the first column in table 3. I suggest ‘Variables / categories’

2. Redo the first and second rows of Table 5 as shown below.

3. Redo the first and second rows of Table 6 as shown below

The author can actually lift the Tables (Tables 5 and 6) as presented here. I have only adjusted the rows for the author. I did not touch the data.

The manuscript is recommended for publication.

Thanks.

Dr. Chukwudi Michael Egbuche

REVIEWER 2:

The manuscript presents data on drivers of LLIN in malaria endemic states in Nigeria. The data is sound and the implication of the study is genuine. However, I think the manuscript should be reread and language should be looked into seriously. Also, the conclusion and implication part at the end should be rewritten to communicate aptly the message in the write up. The abstract can also be improved on. I recommend the paper for publication because it presents information that can be impactful in assisting Nigeria and international funders to focus on drivers of malaria prevalence in the country.

Line 29-30: "The 13 study States were stratified into two based on whether they had increased or reduced parasitaemia between 2015 and 2018.". THIS STATEMENT NEED TO RECASTED FOR CLARITY.

Line 38: "were also more likely of LLIN use". RECAST TO "were also more likely to use LLIN".

Line 43: "were more likely of parasitaemia". RECAST "were more likely to have the parasite".

Line 47-48: "The key drivers of LLIN utilisation mainly related to net source and socioeconomic characteristics which is a key factor for parasitaemia". ALL THE CONCLUSION SHOUKD BE REWRITTEN.

Line 60: World Malaria Report of 2021 should be used instead.

Line 69: The most predominant in Nigeria only or in all regions of the world.

Line 72: I dont think the word "NATIONAL" is needed in the statement "the national prevalence of malaria".

Line 89: Remove the word "THE" in the sentence "in the 13 Nigerian States with"

Line 103: The spelling for "FOCUSED" isnt correct.

Line 216: CHILDREN'S

Line 381: Which other studies?

Line 381-384: The statement need to be recasted

Line 396-398: It has been shown scientifically that children in areas of high malaria transmission intensity develop age398 related immunity. ANY REFERENCE FOR THIS

Line 411: Remove "THE" in the statement "management and the use of antimalarials".

Line 469-475: I think this statement should be removed because this isnt part of conclusion.

Line 489: I think the authors should rewrite the whole statement under "IMPLICATIONS OF FINDINGS FOR POLICY AND PROGRAMMES" as the statements looks more of conclusion.

Reviewers' comments:

Reviewer's Responses to Questions

**Comments to the Author**

1. If the authors have adequately addressed your comments raised in a previous round of review and you feel that this manuscript is now acceptable for publication, you may indicate that here to bypass the “Comments to the Author” section, enter your conflict of interest statement in the “Confidential to Editor” section, and submit your "Accept" recommendation.

Reviewer #1: All comments have been addressed

Reviewer #2: (No Response)

2. Is the manuscript technically sound, and do the data support the conclusions?

Reviewer #1: Yes

Reviewer #2: Yes

3. Has the statistical analysis been performed appropriately and rigorously? 

Reviewer #1: Yes

Reviewer #2: Yes

4. Have the authors made all data underlying the findings in their manuscript fully available?

Reviewer #1: Yes

Reviewer #2: Yes

5. Is the manuscript presented in an intelligible fashion and written in standard English?

Reviewer #1: Yes

Reviewer #2: Yes

6. Review Comments to the Author

Reviewer #1: The author has effected all the corrections pointed out.

Aside that, I noted the following minor corrections which I did not point out the first time (Please I apologize for that).

1. Add title to the first column in table 3. I suggest ‘Variables / categories’

2. Redo the first and second rows of Table 5 as shown below.

3. Redo the first and second rows of Table 6 as shown below

The author can actually lift the Tables (Tables 5 and 6) in the attached document. I have only adjusted the rows for the author. I did not touch the data.

The manuscript is recommended for publication.

Thanks.

Dr. Chukwudi Michael Egbuche

Reviewer #2: The manuscript presents data on drivers of LLIN in malaria endemic states in Nigeria. The data is sound and the implication of the study is genuine. However, I think the manuscript should be reread and language should be looked into seriously. Also, the conclusion and implication part at the end should be rewritten to communicate aptly the message in the write up. The abstract can also be improved on. I recommend the paper for publication because it presents information that can be impactful in assisting Nigeria and international funders to focus on drivers of malaria prevalence in the country.

Line 29-30: "The 13 study States were stratified into two based on whether they had increased or reduced parasitaemia between 2015 and 2018.". THIS STATEMENT NEED TO RECASTED FOR CLARITY.

Line 38: "were also more likely of LLIN use". RECAST TO "were also more likely to use LLIN".

Line 43: "were more likely of parasitaemia". RECAST "were more likely to have the parasite".

Line 47-48: "The key drivers of LLIN utilisation mainly related to net source and socioeconomic characteristics which is a key factor for parasitaemia". ALL THE CONCLUSION SHOUKD BE REWRITTEN.

Line 60: World Malaria Report of 2021 should be used instead.

Line 69: The most predominant in Nigeria only or in all regions of the world.

Line 72: I dont think the word "NATIONAL" is needed in the statement "the national prevalence of malaria".

Line 89: Remove the word "THE" in the sentence "in the 13 Nigerian States with"

Line 103: The spelling for "FOCUSED" isnt correct.

Line 216: CHILDREN'S

Line 381: Which other studies?

Line 381-384: The statement need to be recasted

Line 396-398: It has been shown scientifically that children in areas of high malaria transmission intensity develop age398 related immunity. ANY REFERENCE FOR THIS

Line 411: Remove "THE" in the statement "management and the use of antimalarials".

Line 469-475: I think this statement should be removed because this isnt part of conclusion.

Line 489: I think the authors should rewrite the whole statement under "IMPLICATIONS OF FINDINGS FOR POLICY AND PROGRAMMES" as the statements looks more of conclusion.

7. PLOS authors have the option to publish the peer review history of their article (what does this mean?). If published, this will include your full peer review and any attached files.

Reviewer #1: **Yes: **Dr. Chukwudi Michael Egbuche

Reviewer #2: No

---

## [Author Response · Author response to Decision Letter 1]

6 Apr 2022

COMMENTS TO AUTHOR(S)

The manuscript is sound but there is need for minor revision in some components as elaborated in the comments by the reviewers.

REVIEWER 1:

COMMENTS

The author has effected all the corrections pointed out.

Aside that, I noted the following minor corrections which I did not point out the first time (Please I apologize for that).

1. Add title to the first column in table 3. I suggest ‘Variables / categories’

2. Redo the first and second rows of Table 5 as shown below.

3. Redo the first and second rows of Table 6 as shown below

The author can actually lift the Tables (Tables 5 and 6) as presented here. I have only adjusted the rows for the author. I did not touch the data.

The manuscript is recommended for publication.

Thanks.

Dr. Chukwudi Michael Egbuche

RESPONSE: THANK YOU FOR THE SUGGESTIONS. ALL HAS BEEN IMPLEMENTED

REVIEWER 2:

The manuscript presents data on drivers of LLIN in malaria endemic states in Nigeria. The data is sound and the implication of the study is genuine. However, I think the manuscript should be reread and language should be looked into seriously. Also, the conclusion and implication part at the end should be rewritten to communicate aptly the message in the write up. The abstract can also be improved on. I recommend the paper for publication because it presents information that can be impactful in assisting Nigeria and international funders to focus on drivers of malaria prevalence in the country.

THANK YOU FOR THE CONSTRUCTIVE SUGGESTIONS

Line 29-30: "The 13 study States were stratified into two based on whether they had increased or reduced parasitaemia between 2015 and 2018.". THIS STATEMENT NEED TO RECASTED FOR CLARITY.

RESPONSE: IT HAS BEEN RECASTED

Line 38: "were also more likely of LLIN use". RECAST TO "were also more likely to use LLIN".

RESPONSE: DONE

Line 43: "were more likely of parasitaemia". RECAST "were more likely to have the parasite".

RESPONSE: REVISED AS SUGGESTED

Line 47-48: "The key drivers of LLIN utilisation mainly related to net source and socioeconomic characteristics which is a key factor for parasitaemia". ALL THE CONCLUSION SHOUKD BE REWRITTEN.

RESPONSE: REVISED AS SUGGESTED

Line 60: World Malaria Report of 2021 should be used instead.

THANK YOU. THIS WAS NOT YET OUT AS AT SUBMISSION. WE HAVE USED THE 2021 REPORT NOW

Line 69: The most predominant in Nigeria only or in all regions of the world.

RESPONSE: WE HAVE INDICATED NIGERIA

Line 72: I dont think the word "NATIONAL" is needed in the statement "the national prevalence of malaria".

RESPONSE: REVISED AS SUGGESTED. THANK YOU

Line 89: Remove the word "THE" in the sentence "in the 13 Nigerian States with"

RESPONSE: DONE

Line 103: The spelling for "FOCUSED" isnt correct.

RESPONSE: THANK YOU. IT’S BEEN CORRECTED

Line 216: CHILDREN'S

RESPONSE: CORRECTED

Line 381: Which other studies?

RESPONSE: PREVIOUS STUDIES. REFERENCES PROVIDED

Line 381-384: The statement need to be recasted

RESPONSE: REVISED

Line 396-398: It has been shown scientifically that children in areas of high malaria transmission intensity develop age398 related immunity. ANY REFERENCE FOR THIS

RESPONSE: REFERENCE PROVIDED

Line 411: Remove "THE" in the statement "management and the use of antimalarials".

RESPONSE: THANK YOU. REVISED AS RECOMMENDED

Line 469-475: I think this statement should be removed because this isnt part of conclusion.

RESPONSE: STATEMENT HAS BEEN MOVED TO DISCUSSION SECTION. THANK YOU

Line 489: I think the authors should rewrite the whole statement under "IMPLICATIONS OF FINDINGS FOR POLICY AND PROGRAMMES" as the statements looks more of conclusion.

RESPONSE: NECESSARY REVISIONS HAS ALSO BEEN DONE IN THIS SECTION. THOUGH, IT WAS NOT A CONCLUSION

---

## [Decision Letter · Decision Letter 2]

25 Apr 2022

Drivers of long-lasting insecticide-treated net utilisation and parasitaemia among under-five children in 13 States with high malaria burden in Nigeria

PONE-D-21-27138R2

Dear Dr. Akinyemi,

We’re pleased to inform you that your manuscript has been judged scientifically suitable for publication and will be formally accepted for publication once it meets all outstanding technical requirements.

Kind regards,

Clement Ameh Yaro, Ph.D

Academic Editor

PLOS ONE

Additional Editor Comments (optional):

The author has addressed all the corrections earlier pointed out EXCEPT ONE of those I noted in my second review.

1. Add title to the first column in table 3. I suggested ‘Variables / categories’

You have already done that in Table 2, so repeat same in Table 3.

Reviewers' comments:

Reviewer's Responses to Questions

**Comments to the Author**

1. If the authors have adequately addressed your comments raised in a previous round of review and you feel that this manuscript is now acceptable for publication, you may indicate that here to bypass the “Comments to the Author” section, enter your conflict of interest statement in the “Confidential to Editor” section, and submit your "Accept" recommendation.

Reviewer #1: All comments have been addressed

Reviewer #2: All comments have been addressed

2. Is the manuscript technically sound, and do the data support the conclusions?

Reviewer #1: Yes

Reviewer #2: Yes

3. Has the statistical analysis been performed appropriately and rigorously? 

Reviewer #1: Yes

Reviewer #2: Yes

4. Have the authors made all data underlying the findings in their manuscript fully available?

Reviewer #1: Yes

Reviewer #2: Yes

5. Is the manuscript presented in an intelligible fashion and written in standard English?

Reviewer #1: Yes

Reviewer #2: Yes

6. Review Comments to the Author

Reviewer #1: The author has addressed all the corrections earlier pointed out EXCEPT ONE of those I noted in my second review.

1. Add title to the first column in table 3. I suggested ‘Variables / categories’

You have already done that in Table 2, so repeat same in Table 3.

Reviewer #2: All my previous concerns have been addressed. I believe the manuscript is ready for publication. Thank you

7. PLOS authors have the option to publish the peer review history of their article (what does this mean?). If published, this will include your full peer review and any attached files.

Reviewer #1: **Yes: **Dr. Chukwudi Michael Egbuche

Reviewer #2: **Yes: **Omotayo Ahmed Idowu

---

## [Editor Report · Acceptance letter]

28 Apr 2022

PONE-D-21-27138R2 

Drivers of long-lasting insecticide-treated net utilisation and parasitaemia among under-five children in 13 States with high malaria burden in Nigeria 

Dear Dr. Akinyemi:

I'm pleased to inform you that your manuscript has been deemed suitable for publication in PLOS ONE. Congratulations! Your manuscript is now with our production department. 

Kind regards, 

on behalf of

Dr. Clement Ameh Yaro 

Academic Editor

PLOS ONE